# Structural basis of mechano-chemical coupling by the mitotic kinesin KIF14

Matthieu P.M.H. Benoit [1], Ana B. Asenjo [1], Mohammadjavad Paydar[2], Sabin Dhakal[2], Benjamin H. Kwok [2✉] & Hernando Sosa [1✉]

KIF14 is a mitotic kinesin whose malfunction is associated with cerebral and renal developmental defects and several cancers. Like other kinesins, KIF14 couples ATP hydrolysis and microtubule binding to the generation of mechanical work, but the coupling mechanism between these processes is still not fully clear. Here we report 20 high-resolution (2.7–3.9 Å) cryo-electron microscopy KIF14-microtubule structures with complementary functional assays. Analysis procedures were implemented to separate coexisting conformations of microtubule-bound monomeric and dimeric KIF14 constructs. The data provide a comprehensive view of the microtubule and nucleotide induced KIF14 conformational changes. It shows that: 1) microtubule binding, the nucleotide species, and the neck-linker domain govern the transition between three major conformations of the motor domain; 2) an undocked neck-linker prevents the nucleotide-binding pocket to fully close and dampens ATP hydrolysis; 3) 13 neck-linker residues are required to assume a stable docked conformation; 4) the neck-linker position controls the hydrolysis rather than the nucleotide binding step; 5) the two motor domains of KIF14 dimers adopt distinct conformations when bound to the microtubule; and 6) the formation of the two-heads-bound-state introduces structural changes in both motor domains of KIF14 dimers. These observations provide the structural basis for a coordinated chemo-mechanical kinesin translocation model.

[1] Department Physiology and Biophysics, Albert Einstein College of Medicine, New York, NY, USA. [2] Department of Medicine, Institute for Research in Immunology and Cancer, Université de Montréal, Montreal, QC, Canada. ✉email: benjamin.kwok@umontreal.ca; hernando.sosa@einteinmed.org

KIF14 is a microtubule-based motor protein with essential roles during cell division[1–3]. Its overexpression is associated with tumor progression and resistance to anticancer drugs in several cancers for which it is considered an oncogene[4–9]. Mutations in KIF14 are also associated with neural and kidney development defects[10,11]. However, despite the importance of KIF14, its action mechanism as a kinesin motor is still largely unknown.

KIF14 belongs to the kinesin-3 family of motor proteins. Other members of this family include KIF1A and CeUNC-104[12]. Kinesin-3s in general work as microtubule plus-end-directed motors with the ability to make long processive runs when forming dimers[13]. KIF14 is also reported to be a microtubule plus-end-directed motile kinesin and to protect microtubules against depolymerization[14]. As all kinesins, KIF14 possesses a highly conserved catalytic motor or head domain that contains nucleotide and microtubule binding sites, and where ATP hydrolysis is coupled to the generation of mechanical work. The KIF14 molecule is also similar to other microtubule plus-end-directed motile kinesins in having two motor domain joined by a coiled coil dimerization domain located C-terminal to the motor domain and a ~13 residues long region, the neck-linker, connecting the motor and coiled coil domains[15]. KIF14 also contains an N-terminal extension with a protein-regulating cytokinesis 1 binding domain[1,11,14].

Although much is known regarding the mechanism of action of motile kinesins, and in particular of its founding member kinesin-1, most structural information of kinesin-microtubule complexes is still of limited resolution. As a consequence, it is still not fully clear how conformational changes are coupled to distinct steps of the ATP hydrolysis cycle or how the two heads of a kinesin dimer may coordinate their activities.

Here we report the near-atomic resolution structures of twenty KIF14-microtubule complexes corresponding to five distinct KIF14 constructs in four nucleotide conditions mimicking four key steps of the ATP hydrolysis cycle. We also implemented an analysis procedure to separate coexisting conformations of the kinesin motor bound to the microtubule helical assembly. With this procedure, we determined the structure of the two motor domains of KIF14 dimers bound to the microtubule (two-heads-bound state) at high resolution. The data reveal how microtubule binding alters the structure of the KIF14 motor domain, how changes in nucleotide species bound to the catalytic site are

coupled to motility-related conformational changes and how the two motor domains in a KIF14 dimer coordinate their activities. Our structural data provide a comprehensive molecular understanding of how kinesin ATPase cycle is coordinated with plus-end-directed movement along microtubules.

## Results

**Cryo-electron microscopy (cryo-EM) structures of KIF14 motor domain complexes.** To elucidate conformational changes related to the microtubule-bound KIF14 motor domain ATPase cycle, and their possible modulation by the neck-linker or a partner motor domain, we determined the cryo-EM structures of microtubule (MT) complexes of five distinct KIF14 constructs in four nucleotide conditions. Full-length mouse KIF14 is 1674 amino acids long and the core motor domain resides between residues Asn-391 and Leu-735 (Fig. 1a). The five constructs used in this work comprise residues Asn-391 to Asp-772 (K772), Asn-391 to Lys-755 (K755), Asn-391 to Ala-748 (K748), Asn-391 to Asn-743 (K743), and Asn-391 to Leu-735 (K735). Supplementary Figure 1 shows an alignment of the mouse and human amino-acid sequences in the region of interest. All constructs include the core motor domain but differ in how many residues of the neck-linker (Ile-736 to Leu-750) and the dimerization coiled coil domain (Ile-751 to Ala-764) they include. Note that the residue position taken as the start of the neck-linker varies in the literature by a few positions[16,17]. Here, we take KIF14 Ile-736 as the first neck-linker residue, which is analogous to the definition used in refs. [16,18]. KIF14 Ile-736 aligns with kinesin-1 KIF5B Ile-325. The limits of KIF14 CC1 (Ile-751 to Ala-764) were taken as the residues with a higher than 0.99 COILS[19] coiled coil score. The longer K772 construct includes a full neck-linker and the first coiled coil domain CC1. Construct K755 includes the neck-linker and part of the first coiled coil heptad repeat of the CC1 domain. Construct K748 includes 13 neck-linker residues and no part of the CC1 domain. Construct K743 includes 8 neck-linker residues. Construct K735 comprises the core motor domain and no neck-linker residues. Construct K772 with a full CC1 domain migrates faster than the shorter constructs in a size-exclusion chromatography column (Fig. 1b, c). The faster migration of K772 is consistent with this construct forming dimers in solution, as expected from the presence of a coiled coil domain. Structural snapshots in four distinct steps of the ATPase cycle were obtained

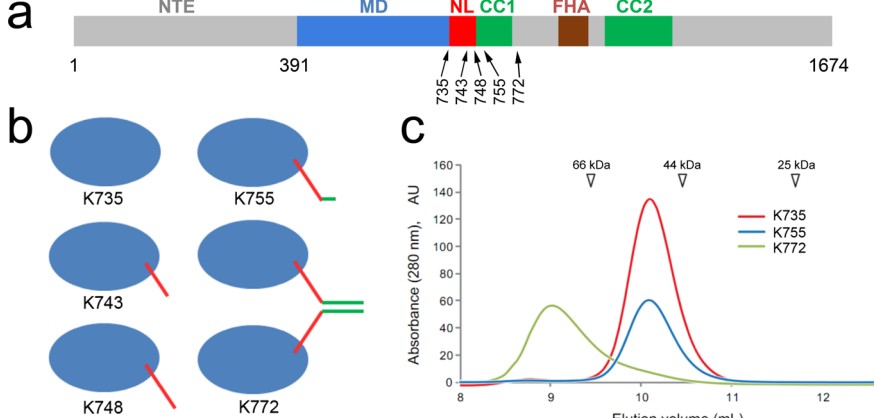

**Fig. 1 KIF14 constructs. a** Primary structure and domain organization of *Mus musculus* KIF14. NTE N-terminal extension, MD motor domain, NL neck-linker domain, FHA Forkhead-associated domain, CC1 coiled coil domain 1, CC2 coiled coil domain 2. Five constructs were used in this work. The constructs include the motor domain from residue 391 at the N-terminal end and up to residues 735, 743, 748, 755, or 772 (relative size of bars and arrow positions not at scale). **b** Predicted quaternary structure of the KIF14 constructs used. **c** Size-exclusion chromatography elution profile of constructs K735, K755, and K772. The migration peak positions of protein standards are indicated (Bovine Serum Albumin, 66 kDa; Ovalbumin, 44 kDa; and Chymotrypsin, 25 kDa).

by adding or removing nucleotides from the experimental solution as follows: (1) ADP added (4 mM) to induce the ADP-bound state, (2) traces of ATP and ADP removed with apyrase to induce the Apo state, (3) non-hydrolysable ATP analog AMP-PNP added (4 mM) to mimic the ATP-bound state, (4) ADP and aluminum fluoride added (4-mM ADP, 2-mM AlCl₃, and 10-mM KF) to mimic the ADP-Pi transition state. The four nucleotide conditions will be denoted through the text as ADP, Apo, ANP, and AAF, respectively.

Combining the different KIF14 constructs and nucleotide conditions, we first calculated eighteen 3D cryo-EM maps using a helical reconstruction method where all the asymmetric subunits in the microtubule helical assembly get averaged. Twelve of these cryo-EM maps revealed one of the two alternative conformations of the KIF14 motor domain while the other six revealed a mixture of conformations averaged. To separate the mixed conformations, we applied to all eighteen datasets a procedure (HASRC, Supplementary Figs. 2 and 3) that refines and classify the 3D densities of individual subunits in a helical assembly. The twelve datasets aforementioned produced classes with only one of the two alternative conformations (open or closed) of the KIF14 motor domain, similar to the ones inferred from the corresponding helically averaged cryo-EM maps. The other six datasets resulted in classes with different KIF14 motor domain conformations. Two datasets (MT-K743-ANP and MT-K743-AAF) produced two high-resolution classes with distinct conformations of the KIF14 motor domain. The other four datasets correspond to KIF14 dimers forming a two-heads-bound intermediate with each of the two heads in a distinct conformation (MT-K755-ANP, MT-K755-AAF, MT-K772-ANP, and MT-K772-AAF). In total twenty cryo-EM maps with twenty-four distinct KIF14-motor-head-tubulin-heterodimer complexes were produced (Table 1 and Fig. 2). The HASRC procedure also resulted in a resolution improvement over the helically averaged maps (Table 1). The attained resolution of the cryo-EM maps, in the 2.7–3.9 Å range (Table 1 and Supplementary Fig. 4), represents a significant improvement over previous kinesin–microtubule complex structures and allowed us to: generate atomic models with high coordinate precision for all the conditions investigated (Supplementary Fig. 5), fully trace the polypeptide chains, locate most amino-acid side chains, and identify the nucleotide species present in the nucleotide-binding pocket. We attributed the improved resolution attained to several factors: (1) use of state of the cryo-EM data collection hardware and software; (2) customized analysis procedures; (3) selection of highly regular helical microtubules; (4) the fact that KIF14 binds tightly to microtubule in all nucleotide conditions, including ADP[14].

The twenty-four structures together with the previously solved crystal structure of the microtubule-unbound KIF14 motor domain[14] can be grouped into three main groups based on the structures of the core motor domain: semi-closed, open/open*, and closed (Table 2 and Supplementary Table 1). In the following sections, we describe and discuss the significance of all the KIF14 structures solved and their differences.

**Microtubule binding opens the KIF14 nucleotide-binding pocket enabling nucleotide exchange**. To examine the effect of microtubule binding on the structure of the KIF14 motor domain, we first compared the microtubule-bound cryo-EM structures with the crystal structure of the ADP-bound KIF14 motor domain[14]. The first thing of notice is that the interaction with the microtubule induces order in regions of the motor domain near the interface with the microtubule, consistent with previous structural studies[20–22]. However, which regions become ordered

and to what extent they are resolved differ among distinct kinesin-microtubule or tubulin complex structures.

Different from kinesin-1, where microtubule binding induces elongation of α-helix-4 (KH4)[23], this helix is as long in the KIF14-ADP crystal structure as in the microtubule-bound structures. In the microtubule-unbound KIF14 crystal structure, parts of the kinesin loops 8, 9, and 11 (KL8, KL9, KL11) are not resolved, and thus presumed disordered. On the other hand, all the KIF14 microtubule-bound cryo-EM structures show clear densities along the full length of these regions (Fig. 3a). KL9 and KL11 correspond to the switch loops (SW1 and SW2, respectively). These switches are common structural elements in the related NTPases, kinesins, myosins, and G-proteins, that sense the nucleotide species in the active site[24]. Densities connecting the side chains of the highly conserved Arg in SW1 and Glu in SW2 (KIF14 R604 and E643), indicative of the formation of a salt bridge between these residues, are resolved in the kinesin-microtubule complexes (Supplementary Fig. 6). Formation of this salt bridge between SW1 and SW2 is considered essential for the ATP hydrolysis catalysis mechanism of kinesins and myosins[25,26]. This density is observed in all the KIF14-microtubule complex structures with a resolution of 3.5 Å or better in the kinesin part of the maps (Supplementary Fig. 6 and Table 1), regardless if the motor is in the open or closed conformations. This indicates that for the switch loops to become ordered and for this salt bridge to form, only microtubule binding is required and not a transition of the core motor domain structure between the open and closed conformations.

Comparing the microtubule-bound and microtubule-unbound structures of KIF14 in the same ADP-bound nucleotide state also reveals a displacement between the tubulin interacting regions (Fig. 3b, c and Supplementary Movie 1). KIF14 helix-6 (KH6) moves relative to helix-4 (KH4), and KIF14 loops 8 and 12 (KL8, KL12) (Fig. 3b–d). The displacement between these areas causes a movement between other regions of the KIF14 motor domain that to a first approximation can be described as a relative rotation between two subdomains within the KIF14 head, a plus-subdomain that includes regions that interact with the microtubule toward the plus end, and a minus subdomain that includes regions that interact with the microtubule toward the minus end (Supplementary Fig. 7). The rotation between these subdomains is most clearly appreciated by the relatively large displacement of regions located at high radius from the rotation axis such as KIF14 α-helix-0 (KH0, Fig. 3b, c). The plus-subdomain includes KIF14 loops KL7, KL8, KL12, α-helices KH4, KH5, and parts of the central β-sheet closer to the microtubule. The minus subdomain includes KIF14 α-helices KH0, KH1, KH2, KH3, KH6, loops KL1, KL2, KL3, KL4 (the P-loop), KL5, and regions of the central β-sheet further away from the microtubule. The switch loops (KL9 and KL11) can be considered a third subdomain to account for the fact that they become ordered with microtubule binding and move in a different direction from the other two subdomains depending on the nucleotide present in the active site (see next section). Because the kinesin nucleotide-binding pocket is located between the three subdomains, relative movement between them results in an altered nucleotide-binding pocket architecture (Fig. 3d and Supplementary Fig. 7c). From the KIF14 microtubule-unbound to the microtubule-bound structure, the distance between the switch loops and the P-loop increases and there is a large displacement in the position of KH0 relative to the microtubule and the switch loops (Fig. 3d and Table 2). These structural rearrangements disrupt interactions between the ADP phosphate groups, the switch regions, and the P-loop providing an explanation for the acceleration of product release associated

**Table 1 Cryo-EM data collection, refinement, and validation statistics.**

| | K735-ANP (EMDB-21949) (PDB 6WWV) | K735-AAF (EMDB-21948) (PDB 6WWU) | K735 Apo (EMDB-21947) (PDB 6WWT) | K743-ANP open state class (EMDB-21946) (PDB 6WWS) |
|---|---|---|---|---|
| **Data collection and processing** | | | | |
| Magnification (actual) | 46,598 | 60,168 | 46,598 | 58,893 |
| Voltage (kV) | 300 | 300 | 300 | 300 |
| Electron exposure (e−/Å²) | 69.9 | 70.1 | 69.9 | 63.0 |
| Defocus range (μm)[a] | 1.0–1.9 | 1.1–1.9 | 1.0–2.0 | 0.9–1.6 |
| Pixel size (Å) | 1.073 | 0.831 | 1.073 | 0.849 |
| Symmetry imposed[b] | Helical | Helical | Helical | Helical |
| Rise (Å) | 5.5 | 5.41 | 5.46 | 5.45 |
| Twist (deg) | 168.09 | 168.07 | 168.09 | 168.09 |
| Particle images identified as 15R symmetry (no.) | 15,350 | 18,849 | 9865 | 38,121 |
| Particle images in helical reconstruction (no.) | 15,350 | 18,849 | 9865 | 38,121 |
| Single particles (no.)[c] | 230,250 | 282,735 | 147,975 | 571,815 |
| Single particles used (no.)[d] | 139,592 | 217,725 | 101,569 | 324,809 |
| Overall resolution (Å) | 3.1 | 2.7 | 3.2 | 2.7 |
| FSC threshold | 0.143 | 0.143 | 0.143 | 0.143 |
| Kinesin resolution (Å) | 3.2 | 2.9 | 3.3 | 2.8 |
| Tubulin resolution (Å) | 3.0 | 2.7 | 3.2 | 2.6 |
| Helical resolution (Å)[e] | 3.5 | 3.0 | 3.6 | 3.0 |
| **Refinement** | | | | |
| Initial model used (PDB code)[f] | 6B0I | 6B0I | 6B0I | 6B0I |
| **Model composition** | | | | |
| Non-hydrogen atoms | 9676 | 9688 | 9657 | 9697 |
| Protein residues | 1215 | 1216 | 1215 | 1216 |
| Ligands | 5 | 5 | 4 | 5 |
| **R.m.s. deviations** | | | | |
| Bond lengths (Å) | 0.008 | 0.0101 | 0.0083 | 0.0090 |
| Bond angles (°) | 1.21 | 1.42 | 1.43 | 1.42 |
| **Validation** | | | | |
| MolProbity score | 1.74 | 2.12 | 1.92 | 2.25 |
| Clashscore | 5.67 | 6.82 | 6.00 | 8.07 |
| Poor rotamers (%) | 0.67 | 2.78 | 1.53 | 3.06 |
| **Ramachandran plot** | | | | |
| Favored (%) | 93.47 | 93.97 | 92.89 | 93.14 |
| Allowed (%) | 5.96 | 5.79 | 6.78 | 6.53 |
| Disallowed (%) | 0.58 | 0.25 | 0.33 | 0.33 |

| | K743-ANP closed state class (EMDB-23540) (PDB 7LVQ) | K743-AAF open state class (EMDB-21945) (PDB 6WWR) | K743-AAF closed state class (EMDB-23541) (PDB 7LVR) | K743-ADP (EMDB-21944) (PDB 6WWQ) |
|---|---|---|---|---|
| **Data collection and processing** | | | | |
| Magnification (actual) | 58,893 | 60,606 | 60,606 | 60,386 |
| Voltage (kV) | 300 | 300 | 300 | 300 |
| Electron exposure (e−/Å²) | 63.0 | 70.8 | 70.8 | 70.2 |
| Defocus range (μm)[a] | 0.9–1.6 | 0.8–1.7 | 0.8–1.7 | 0.8–1.6 |
| Pixel size (Å) | 0.849 | 0.825 | 0.825 | 0.828 |
| Symmetry imposed[b] | Helical | Helical | Helical | Helical |
| Rise (Å) | 5.45 | 5.44 | 5.44 | 5.42 |
| Twist (deg) | 168.09 | 168.07 | 168.07 | 168.08 |
| Particle images identified as 15 R symmetry (no.) | 38,121 | 41,714 | 41,714 | 5194 |
| Particle images in helical reconstruction (no.) | 5194 | 41,714 | 41,714 | 5194 |
| Single particles (no.)[c] | 571,815 | 348,209 | 625,710 | 77,910 |
| Single particles used (no.)[d] | 136,802 | 348,209 | 162,897 | 61,697 |
| Overall resolution (Å) | 2.9 | 2.7 | 2.9 | 3.0 |
| FSC threshold | 0.143 | 0.143 | 0.143 | 0.143 |
| Kinesin resolution (Å) | 3.1 | 2.9 | 3.1 | 3.3 |
| Tubulin resolution (Å) | 2.8 | 2.6 | 2.7 | 3.0 |
| Helical resolution (Å)[e] | 3.0 | 3.0 | 3.0 | 3.3 |
| **Refinement** | | | | |
| Initial model used (PDB code)[f] | 6B0I | 6B0I | 6B0I | 6B0I |
| Model composition | | | | |

**Table 1 (continued)**

|  | K743-ANP closed state class (EMDB-23540) (PDB 7LVQ) | K743-AAF open state class (EMDB-21945) (PDB 6WWR) | K743-AAF closed state class (EMDB-23541) (PDB 7LVR) | K743-ADP (EMDB-21944) (PDB 6WWQ) |
|---|---|---|---|---|
| Non-hydrogen atoms | 9768 | 9719 | 9802 | 9675 |
| Protein residues | 1226 | 1220 | 1231 | 1215 |
| Ligands | 6 | 5 | 7 | 5 |
| R.m.s. deviations |  |  |  |  |
| Bond lengths (Å) | 0.0237 | 0.0089 | 0.0126 | 0.0087 |
| Bond angles (°) | 1.48 | 1.19 | 1.25 | 1.19 |
| Validation |  |  |  |  |
| MolProbity score | 1.71 | 1.68 | 1.53 | 1.67 |
| Clashscore | 5.31 | 5.29 | 3.79 | 5.15 |
| Poor rotamers (%) | 0.66 | 0.57 | 0.19 | 0.19 |
| Ramachandran plot |  |  |  |  |
| Favored (%) | 93.52 | 94.15 | 94.78 | 94.13 |
| Allowed (%) | 6.39 | 5.85 | 5.22 | 5.87 |
| Disallowed (%) | 0.08 | 0.00 | 0.00 | 0.00 |

|  | K743 Apo (EMDB-21943) (PDB 6WWP) | K748-ANP (EMDB-21942) (PDB 6WWO) | K748-AAF (EMDB-21941) (PDB 6WWN) | K748-ADP (EMDB-21940) (PDB 6WWM) |
|---|---|---|---|---|
| Data collection and processing |  |  |  |  |
| Magnification (actual) | 58,962 | 60,386 | 60,606 | 58,893 |
| Voltage (kV) | 300 | 300 | 300 | 300 |
| Electron exposure (e⁻/Å²) | 70.0 | 68.2 | 70.5 | 62.4 |
| Defocus range (μm)[a] | 0.9–1.8 | 0.9–1.7 | 1.0–1.7 | 0.8–1.9 |
| Pixel size (Å) | 0.848 | 0.828 | 0.825 | 0.825 |
| Symmetry imposed[b] | Helical | Helical | Helical | Helical |
| Rise (Å) | 5.45 | 5.43 | 5.41 | 5.51 |
| Twist (deg) | 168.09 | 168.08 | 168.07 | 168.09 |
| Particle images identified as 15 R symmetry (no.) | 12,644 | 16,440 | 15,178 | 14,102 |
| Particle images in helical reconstruction (no.) | 12,644 | 11,075 | 15,178 | 14,102 |
| Single particles (no.)[c] | 189,660 | 246,600 | 227,670 | 211,530 |
| Single particles used (no.)[d] | 147,872 | 166,809 | 161,836 | 111,394 |
| Overall resolution (Å) | 3.1 | 2.8 | 3.5 | 2.8 |
| FSC threshold | 0.143 | 0.143 | 0.143 | 0.143 |
| Kinesin resolution (Å) | 3.3 | 2.9 | 3.5 | 3.0 |
| Tubulin resolution (Å) | 3.0 | 2.8 | 3.4 | 2.7 |
| Helical resolution (Å)[e] | 3.3 | 3.2 | 3.9 | 3.1 |
| Refinement |  |  |  |  |
| Initial model used (PDB code)[f] | 6B0I | 6B0I | 6B0I | 6B0I |
| Model composition |  |  |  |  |
| Non-hydrogen atoms | 9646 | 9784 | 9784 | 9665 |
| Protein residues | 1213 | 1230 | 1230 | 1212 |
| Ligands | 4 | 6 | 7 | 5 |
| R.m.s. deviations |  |  |  |  |
| Bond lengths (Å) | 0.0068 | 0.0102 | 0.0068 | 0.0118 |
| Bond angles (°) | 1.18 | 1.41 | 1.20 | 1.24 |
| Validation |  |  |  |  |
| MolProbity score | 1.80 | 1.98 | 1.78 | 1.73 |
| Clashscore | 6.69 | 5.92 | 6.13 | 5.42 |
| Poor rotamers (%) | 0.67 | 1.99 | 0.66 | 0.77 |
| Ramachandran plot |  |  |  |  |
| Favored (%) | 93.29 | 93.55 | 93.14 | 93.12 |
| Allowed (%) | 6.55 | 6.13 | 6.70 | 6.88 |
| Disallowed (%) | 0.17 | 0.33 | 0.16 | 0.00 |

|  | K755 ANP (EMDB-21939) (PDB 6WWL) | K755-AAF (EMDB-21938) (PDB 6WWK) | K755-ADP (EMDB-21937) (PDB 6WWJ) | K755 Apo (EMDB-21936) (PDB 6WWI) |
|---|---|---|---|---|
| Data collection and processing |  |  |  |  |
| Magnification (actual) | 46,624 | 58,962 | 60,168 | 45,956 |
| Voltage (kV) | 300 | 300 | 300 | 300 |
| Electron exposure (e⁻/Å²) | 72.1 | 73.4 | 70.6 | 69.2 |
| Defocus range (μm)[a] | 0.6–2.0 | 0.6–1.7 | 0.7–1.7 | 0.7–1.8 |

**Table 1 (continued)**

| | K755 ANP (EMDB-21939) (PDB 6WWL) | K755-AAF (EMDB-21938) (PDB 6WWK) | K755-ADP (EMDB-21937) (PDB 6WWJ) | K755 Apo (EMDB-21936) (PDB 6WWI) |
|---|---|---|---|---|
| Pixel size (Å) | 1.0724 | 0.848 | 0.831 | 1.088 |
| Symmetry imposed[b] | Helical | Helical | Helical | Helical |
| Rise (Å) | 5.47 | 5.45 | 5.46 | 5.5 |
| Twist (deg) | 168.09 | 168.07 | 168.09 | 169.09 |
| Particle images identified as 15 R symmetry (no.) | 32,006 | 26,000 | 18,744 | 15,798 |
| Particle images in helical reconstruction (no.) | 32,006 | 26,000 | 18,744 | 15,798 |
| Single particles (no.)[c] | 480,090 | 390,000 | 281,160 | 236,970 |
| Single particles used (no.)[d] | 152,330 | 146,982 | 171,108 | 141,605 |
| Overall resolution (Å) | 3.1 | 3.0 | 3.4 | 3.6 |
| FSC threshold | 0.143 | 0.143 | 0.143 | 0.143 |
| Kinesin resolution (Å) | 3.3 | 3.1 | 3.5 | 3.8 |
| Tubulin resolution (Å) | 3.1 | 2.9 | 3.3 | 3.5 |
| Helical resolution (Å)[e] | 3.4 | 3.2 | 3.5 | 3.8 |
| Refinement | | | | |
| Initial model used (PDB code)[f] | 6B0I | 6B0I | 6B0I | 6B0I |
| Model composition | | | | |
| Non-hydrogen atoms | 196,65 | 19,698 | 9798 | 9753 |
| Protein residues | 2470 | 2473 | 1230 | 1229 |
| Ligands | 11 | 11 | 5 | 4 |
| R.m.s. deviations | | | | |
| Bond lengths (Å) | 0.0072 | 0.0073 | 0.0086 | 0.0110 |
| Bond angles (°) | 1.10 | 1.10 | 1.23 | 1.25 |
| Validation | | | | |
| MolProbity score | 1.49 | 1.71 | 1.80 | 1.84 |
| Clashscore | 4.88 | 4.79 | 6.69 | 6.71 |
| Poor rotamers (%) | 0.94 | 1.65 | 0.57 | 0.47 |
| Ramachandran plot | | | | |
| Favored (%) | 96.42 | 95.77 | 93.38 | 92.56 |
| Allowed (%) | 3.58 | 4.14 | 6.62 | 7.36 |
| Disallowed (%) | 0.00 | 0.08 | 0.00 | 0.08 |

| | K772ANP (EMDB-21935) (PDB 6WWH) | K772-AAF (EMDB-21934) (PDB 6WWG) | K772-ADP (EMDB-21933) (PDB 6WWF) | K772 Apo (EMDB-21932) (PDB 6WWE) |
|---|---|---|---|---|
| Data collection and processing | | | | |
| Magnification (actual) | 46,598 | 60,533 | 58,893 | 58,824 |
| Voltage (kV) | 300 | 300 | 300 | 300 |
| Electron exposure (e$^-$/Å$^2$) | 67.3 | 75.1 | 66.8 | 68.7 |
| Defocus range (μm)[a] | 0.8–2.0 | 0.7–1.6 | 0.5–1.5 | 0.7–1.9 |
| Pixel size (Å) | 1.073 | 0.826 | 0.849 | 0.850 |
| Symmetry imposed[b] | Helical | Helical | Helical | Helical |
| Rise (Å) | 5.48 | 5.44 | 5.48 | 5.50 |
| Twist (deg) | 168.09 | 168.07 | 168.09 | 168.08 |
| Particle images identified as 15 R symmetry (no.) | 3395 | 23,479 | 7377 | 3843 |
| Particle images in helical reconstruction (no.) | 3395 | 23,479 | 7377 | 3843 |
| Single particles (no.)[c] | 50,925 | 352,185 | 110,385 | 57,645 |
| Single particles used (no.)[d] | 21,458 | 135,651 | 73,851 | 42,831 |
| Overall resolution (Å) | 3.8 | 2.9 | 3.3 | 3.9 |
| FSC threshold | 0.143 | 0.143 | 0.143 | 0.143 |
| Kinesin resolution (Å) | 4.3 | 3.1 | 3.5 | 4.3 |
| Tubulin resolution (Å) | 3.6 | 2.8 | 3.2 | 3.8 |
| Helical resolution (Å)[e] | 3.9 | 3.0 | 3.5 | 3.9 |
| Refinement | | | | |
| Initial model used (PDB code)[f] | 6B0I | 6B0I | 6B0I | 6B0I |
| Model composition | | | | |
| Non-hydrogen atoms | 19,685 | 19,749 | 9805 | 9777 |
| Protein residues | 2471 | 2479 | 1231 | 1231 |
| Ligands | 12 | 12 | 5 | 4 |
| R.m.s. deviations | | | | |
| Bond lengths (Å) | 0.0144 | 0.0068 | 0.0072 | 0.0069 |

**Table 1 (continued)**

| | K772ANP (EMDB-21935) (PDB 6WWH) | K772-AAF (EMDB-21934) (PDB 6WWG) | K772-ADP (EMDB-21933) (PDB 6WWF) | K772 Apo (EMDB-21932) (PDB 6WWE) |
|---|---|---|---|---|
| Bond angles (°) | 1.27 | 1.16 | 1.26 | 1.23 |
| Validation | | | | |
| MolProbity score | 2.35 | 2.01 | 1.66 | 1.84 |
| Clashscore | 11.82 | 6.48 | 4.97 | 7.11 |
| Poor rotamers (%) | 2.92 | 2.58 | 0.47 | 0.19 |
| Ramachandran plot | | | | |
| Favored (%) | 93.86 | 95.05 | 94.12 | 92.90 |
| Allowed (%) | 5.98 | 4.66 | 5.88 | 6.94 |
| Disallowed (%) | 0.16 | 0.28 | 0.00 | 0.16 |

[a]Range of the average defocus measured on the particle images used for the helical reconstructions. The range comprises 90% of the particles used (5% of the particles defocus are below and 5% are above this range).
[b]Symmetry imposed on the helically averaged map only.
[c]Total number of particles after symmetry expansion.
[d]Number of single particles used after 3D classification. For single-head-bound states, these single particles correspond to a kinesin motor bound to a tubulin dimer, and for two-head-bound kinesin states (755-ANP, 755-AAF, 772-ANP, and 772-AAF), they correspond to two connected kinesin motors bound to two tubulin dimers (i.e., half the corresponding amount of particles from symmetry expansion).
[e]Overall resolution of the helically averaged map.
[f]15R decorated microtubule model used as the start of cryo-EM processing.

with kinesin microtubule binding[27]. We also propose that the open nucleotide conformation and the large displacement of KH0 relative to the plus subdomain facilitates the incorporation of ATP into the nucleotide-binding pocket and the exchange of ADP for ATP. This is consistent with molecular dynamics simulations that link movements of KH0, KL5, and SW1 with the incorporation of ATP into the nucleotide-binding pocket[28]. Also, consistent with this proposal, we find that ATP analogs can readily bind to the open nucleotide-binding conformation (Figs. 2, 5f, g, 7a–c, and Supplementary Fig. 11g).

All the KIF14 microtubule-bound structures in the ADP or Apo states are very similar with the motor domain adjusting to the open conformation (Fig. 3e, Table 2, and Supplementary Table 1). The twist of the motor domain central β-sheet in all the open structures is also similar to the one found in the semi-closed conformation of the microtubule-unbound KIF14 crystal structure (Fig. 3f and Table 2). This central β-sheet is another structural element in common with related NTPases, and based on comparison of myosin structures, it is thought that a more twisted configuration represents an ADP release/Apo intermediate[14,29]. However, analysis of all the MT-KIF14 complexes reveals that the twist of the central beta-sheet is better correlated with whether the motor domain is in the closed or open conformation, rather than the particular nucleotide species in the active site. All the microtubule-bound KIF14 open structures have the more twisted β-sheet conformation, regardless of the nucleotide species in the active site. The less-twisted central β-sheet is observed only in the closed configurations of the motor domain.

**ATP analogs induce closure of the KIF14 nucleotide-binding pocket**. The ATP analog AMP-PNP and the ADP-Pi analog ADP-AlF$_x$ induce large conformational changes in the KIF14 constructs with at least 13 neck-linker residues (K748, K755, and K772; Table 2, Supplementary Movie 1). Comparing the MT-K748-ANP and MT-K748-AAF with any of the Apo or ADP structures reveals a large rotation (counterclockwise when viewed from the microtubule minus end) of the minus subdomain (Fig. 4a, b and Supplementary Movie 1). The motor domain structure of the AMP-PNP and ADP-AlF$_x$ complexes are very similar, both conforming to the closed configuration (Table 2 and Supplementary Table 1), but they are not identical. We found small but consistent differences between the

ANP and AAF closed motor domain structures (Supplementary Fig. 9). The AAF closed structures are slightly rotated (counterclockwise when viewed from the microtubule minus end) relative to the ANP closed structures. The rotation of the minus subdomain going from the ADP to the ANP or AAF closed structures is in the opposite direction to the one produced by microtubule binding in the Apo or ADP states and larger in magnitude (Fig. 3b, c vs. Fig. 4a, b). It is also accompanied by a reduction in the twist of the central β-sheet (Fig. 3f and Table 2) and an opening of the hydrophobic pocket formed between KIF14 β-strand-1 (KS1) and helix-4 (KH4) that allows the neck-linker to dock onto the motor domain (Fig. 4e, f). The neck-linker is docked onto the motor domain establishing contacts with KS1 and forming the cover-neck bundle, a structural motif thought to stabilize the neck-linker in the docked configuration and to be important for force generation[30,31]. Further contacts at the tip of the motor domain are formed between the neck-linker and β-strands 3 and 7 (Fig. 4e, f).

In addition, the switch loops move toward the nucleotide in a direction opposite to the movement of the minus subdomain to establish a more closed nucleotide-binding pocket (Fig. 4c, d). There are also changes in the density associated with the tips of the switch loops that indicate changes in the mobility of these regions (Fig. 4g, h). In the ADP structures, the density at the tip of SW1 is relatively weak indicating higher mobility in this area compared to the tip of SW2. This situation is reversed in the ANP closed structures.

The rotation of the minus subdomain and the movement of the switch loops from the Apo/ADP to the ANP/AAF closed structures is accompanied by structural rearrangements at the microtubule-KIF14 interface. KH6 and the SW2 loop move relative to the microtubule (Fig. 4b, i, j). This rearrangement disrupts interactions between KH6 and SW2 with α-tubulin (Supplementary Fig. 8). A salt bridge between KIF14 Arg-728 in KH6 and α-tubulin Glu-423 is disrupted and the side chain of α-tubulin Tyr-108 changes from a mixture of alternate rotamer conformations to a single one (Fig. 4i, j). As in the MT-K748-ADP/Apo structures (see previous section), side chain densities suggesting the formation of a salt bridge between Arg-604 in SW1 and Glu-643 in SW2 are observed (Supplementary Fig. 6). The switch loops, except the differences indicated above at their tips and near the microtubule interface, have similar topology in all microtubule-bound structures.

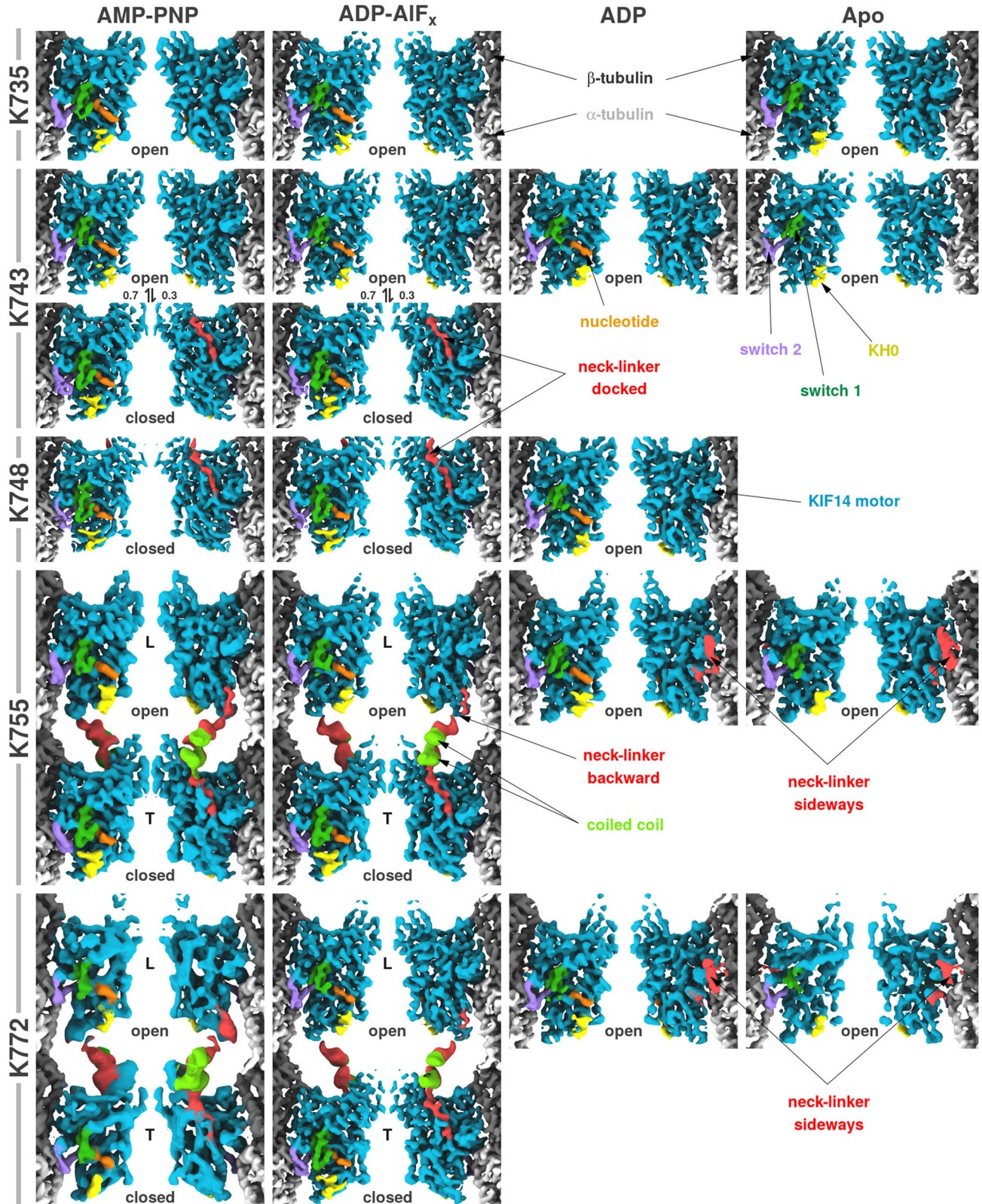

The configuration of the motor domain around the nucleotide phosphate groups in the MT-K748-ANP/AAF structures is similar to the closed ATP-hydrolysis-catalytic configuration, first observed in the crystal structure of the kinesin-5 EG5 in the presence of AMP-PNP[25]. This conformation of the KIF14 motor domain, together with the KIF14 Apo/ADP microtubule complexes and the KIF14-ADP crystal structure, define the three major conformations of the KIF14 motor domain that we named,

**Fig. 2 Cryo-EM 3D density maps.** Iso-density surface representations of the twenty cryo-EM maps determined. The columns correspond to the four different nucleotide conditions used and the rows to the five KIF14 constructs used. Each panel shows two views of the MT-KIF14 3D map rotated 180° from each other (MT protofilament partially shown). Surfaces are colored according to the fitted atomic structure they enclose: α-tubulin in light gray, β-tubulin in dark gray, most of the kinesin motor domain in blue with the switch-1 loop in green, the switch-2 loop in magenta, KH0 in yellow, the bound nucleotide (ADP or AMP-PNP) in orange, the neck-linker in red, and the CC1 coiled coil in bright green. Closed or open refers to the deduced conformation of the motor domain. T and L indicate trailing or leading motor domains in the two-heads-bound dimer structures. Numbers between the K743 AMP-PNP and ADP-AlFx, open and closed structures indicate the relative population of each conformation (open or closed). Densities in the undocked neck-linker and coiled coil (only present in the K755 and K772 maps) are noisier than the rest of the map and are displayed at a lower contour level after low pass-filtration to 7–8 Å. The figure was prepared with VMD[69].

closed, open, and semi-closed based on the structure of the nucleotide-binding pocket (Table 2, Supplementary Table 1, and Supplementary Movie 1).

**Full neck-linker docking is required for complete nucleotide-binding pocket closure.** Most models for the coordinated movement of kinesins along microtubules couple some step of the ATPase cycle with the position of the neck-linker domain[32,33]. However, how the position of the neck-linker may control the ATPase cycle is unclear. There is no direct structural data for any kinesin indicating how the position of the neck-linker could affect the structure of the nucleotide-binding pocket. To investigate this issue, we obtained the cryo-EM structures of microtubules in complex with the constructs K735 and K743 that have truncated neck-linkers (Fig. 1a, b). We surmised that an absent or truncated neck-linker would mimic the situation when the neck-linker is mechanically pulled out from the docked configuration as it may occur under force or in a two-heads microtubule-bound intermediate.

In the Apo and ADP states the motor domain structures are very similar for all the complexes investigated; this is regardless of whether the KIF14 construct contained a truncated or a full neck-linker, they all conform to the open configuration (Fig. 3e and Table 2). On the other hand, in the presence of AMP-PNP or ADP-AlF$_x$, the motor domain structures of the MT-K735 and MT-K743 complexes do not completely resemble the closed configuration observed with constructs that include a full neck-linker (Fig. 2, Table 2, and Supplementary Table 1).

The motor domain in the MT-K735-ANP complex shows a new configuration where KH0 is at an intermediate position between the open and closed configuration (Fig. 5a), but as in the open configuration the switch loops remain further away from the P-loop (Fig. 5b). In the MT-K735-AAF structure, the motor domain closely resembles the open conformation observed in the Apo/ADP states (Fig. 5a, b). These results show that without the neck-linker: (1) ATP analogs with the γ-phosphate group covalently linked such as AMP-PNP induce a partial rotation of the minus subdomain toward the closed configuration, (2) the post-hydrolysis analog ADP-AlF$_x$ does not induce any subdomain rotation and the motor domain resembles the open configuration, (3) neither analog induces the closed configuration of the motor domain observed with the constructs that include at least 13 neck-linker residues.

The structures of the MT-K743 complex in the AMP-PNP and ADP-AlF$_x$ also do not fully conform to the closed configuration. The helically averaged cryo-EM maps are best-fitted with an open configuration KIF14 model, but different from the MT-K735 complexes, these maps also include weaker densities corresponding to the closed configuration (Supplementary Fig. 10e). The 3D classification procedure resulted in two distinct high-resolution classes corresponding to the open and closed configuration with the majority of the data (~70% of particles in the two classes) corresponding to the open conformation class. These cryo-EM

maps and their corresponding fitted atomic models were named: MT-K743-ANP-O, MT-K743-AAF-O, MT-K743-ANP-C, and MT-K743-AAF-C, where the suffix O and C denotes the open and closed conformations respectively. Similar to other open-conformation structures, the more represented MT-K743-ANP-O and MT-K743-AAF-O structures do not show a resolved docked neck-linker. The lack of a docked neck-linker in the majority of the population is surprising considering that the K743 construct contains the residues that form the cover-neck bundle and the asparagine latch (N743 in KIF14), thought to hold the neck-linker in the docked conformation[30,34]. The fact that the K743 construct cannot adopt a stable docked configuration but the K748 can, shows that the additional contacts formed by the extra five residues of the K748 construct (Fig. 4f) are required to stabilize the neck-linker docked configuration.

Taken together the results with the K735 and K743 constructs show that without a fully docked neck-linker a single KIF14 motor domain cannot adopt a stable closed catalytic conformation and instead adopt a conformation most similar to the open structures observed in the Apo and ADP states. These structures can be grouped into a subgroup, open* (Table 2 and Supplementary Table 1) indicative of the fact that they are open conformations despite being in the presence of nucleotides that induce the closed conformation and that they lack a docked neck-linker. Within this group, the most distinctive conformation of the motor domain corresponds to the intermediate configuration observed in the MT-K735-ANP complex (Fig. 5a, b, Table 2, and Supplementary Table 1).

Consistent with the structural results, we found that the K735 and K743 constructs have reduced microtubule-stimulated ATPase activity relative to the longer KIF14 constructs (Fig. 5c). This is also consistent with previous reports indicating a large reduction of the microtubule-stimulated ATPase activity of other kinesins with truncated neck-linkers[18,35]. Thus, the results show that the neck-linker position and the ATPase activity of the motor domain are reciprocally coupled. ATP binding induces closure of the nucleotide-binding pocket and neck-linker docking (as shown in the MT-K748-ANP structure). Conversely, preventing neck-linker docking impedes full closure of the nucleotide-binding pocket and inhibits the ATPase activity.

**Unique ligands position in the open* motor domain conformation.** The open* conformations MT-K735-ANP/AAF and MT-K743-O-ANP/AAF cryo-EM maps show clear nucleotide-associated densities in the active site indicating that both nucleotide analogs bind to the open* configuration of the motor domain (Fig. 2, Fig. 5e–g, and Supplementary Fig. 11d, f, g). In the case of the ANP open* structures, AMP-PNP maintains a similar location relative to the P-loop and KL5 than in the ANP closed structures, but it is further away from the switch loops. This disrupts interactions between the nucleotide phosphate groups and residues in switch-1 that are observed in the closed ANP structures, including the coordination between the γ-phosphate group, the Mg$^{2+}$ ion, and switch-1 Ser-603.

**Table 2 KIF14 conformation groups key features.**

| | Nucleotide | MT-bound | Selected NBP distances (Å) | | | Neck-linker | Model | PDB ID |
|---|---|---|---|---|---|---|---|---|
| | | | K488-R604 (SW1-PL) | R406-T647 (KHO-SW2) | Cβ-twist KS3∠KS4 | | | |
| Semi-Closed | ADP | No | **13.4** | **21.4** | **45.8** | Undocked, mobile | K-ADP | 4OZQ |
| Open | ADP | Yes | 14.3 | 24.9 | 44.6 | Undocked, mobile | MT-K743-ADP | 6WWQ |
| | ADP | Yes | 14.8 | 23.7 | 42.8 | Undocked, mobile | MT-K748-ADP | 6WWM |
| | ADP | Yes | 14.4 | 24.5 | 46.0 | Undocked, sideways | MT-K755-ADP | 6WWJ |
| | ADP | Yes | 14.8 | 22.4 | 42.2 | Undocked, sideways | MT-K772-ADP | 6WWF |
| | Apo | Yes | 14.7 | 23.0 | 45.4 | Not present | MT-K735-Apo | 6WWT |
| | Apo | Yes | 14.1 | 23.6 | 42.7 | Undocked, mobile | MT-K743-Apo | 6WWP |
| | Apo | Yes | 14.6 | 24.9 | 44.4 | Undocked, sideways | MT-K755-Apo | 6WWI |
| | Apo | Yes | 15.2 | 25.4 | 41.8 | Undocked, sideways | MT-K772-Apo | 6WWE |
| | | **Avg** | **14.6** | **24.0** | **43.7** | | | |
| | | **sd** | **0.3** | **1.1** | **1.6** | | | |
| Open* | AMP-PNP | Yes | 14.2 | 20.9 | 44.0 | Not present | MT-K735-ANP | 6WWV |
| | AMP-PNP | Yes | 14.3 | 24.2 | 45.6 | Undocked, mobile | MT-K743-ANP-O | 6WWS |
| | AMP-PNP | Yes | 14.3 | 24.4 | 44.1 | Undocked, backwards | MT-K755-ANP-L | 6WWL |
| | AMP-PNP | Yes | 14.0 | 22.2 | 43.3 | Undocked, backwards | MT-K772-ANP-L | 6WWH |
| | ADP-AIF$_X$ | Yes | 14.4 | 23.5 | 42.6 | Not present | MT-K735-AAF | 6WWU |
| | ADP-AIF$_X$ | Yes | 14.3 | 23.8 | 45.1 | Undocked, mobile | MT-K743-AAF-O | 6WWR |
| | ADP-AIF$_X$ | Yes | 14.8 | 22.8 | 44.0 | Undocked, backwards | MT-K755-AAF-L | 6WWK |
| | ADP-AIF$_X$ | Yes | 14.5 | 21.8 | 44.7 | Undocked, backwards | MT-K772-AAF-L | 6WWG |
| | | **Avg** | **14.3** | **22.9** | **44.2** | | | |
| | | **sd** | **0.2** | **1.2** | **1.0** | | | |
| Closed | AMP-PNP | Yes | 12.7 | 12.0 | 36.4 | Docked[a] | MT-K743-ANP-C | 7LVQ |
| | AMP-PNP | Yes | 12.8 | 11.2 | 37.2 | Docked | MT-K748-ANP | 6WWO |
| | AMP-PNP | Yes | 12.3 | 11.0 | 36.4 | Docked, partially unzipped | MT-K755-ANP-T | 6WWL |
| | AMP-PNP | Yes | 12.3 | 13.0 | 35.0 | Docked, partially unzipped | MT-K772-ANP-T | 6WWH |
| | ADP-AIF$_X$ | Yes | 13.1 | 11.2 | 33.7 | Docked[a] | MT-K743-AAF-C | 7LVR |
| | ADP-AIF$_X$ | Yes | 12.4 | 11.6 | 35.4 | Docked | MT-K748-AAF | 6WWN |
| | ADP-AIF$_X$ | Yes | 11.9 | 11.9 | 37.0 | Docked, partially unzipped | MT-K755-AAF-T | 6WWK |
| | ADP-AIF$_X$ | Yes | 12.8 | 11.4 | 36.7 | Docked, partially unzipped | MT-K772-AAF-T | 6WWG |
| | | **Avg** | **12.5** | **11.7** | **36.0** | | | |
| | | **sd** | **0.4** | **0.6** | **1.2** | | | |

Selected nucleotide-binding pocket (NBP) distances correspond to the distance between Cα carbons in selected residues around the NBP. Central β-sheet twist (Cβ-twist) was measured as the angle between two vectors formed by the coordinates of the Cα carbons of KIF14 residues 475 and 480 in KS3 and residues 527 and 531 in KS4 (Fig. 3f). L and T indicate respectively the leading or the trailing head in microtubule two-heads-bound structures. Average (avg) and standard deviation (sd) values of the group of structures in the rows above are shown in bold.

KHO kinesin α-helix-0, KS3 kinesin β-strand 3, KS4 kinesins β-strand 4, PL P-loop, SW1 switch-1 loop, SW2 switch-2 loop.

ᵃConstruct K743 has the neck-linker truncated at position 743. A minority of the MT-K743-ANP/AAF data (~30%) is associated with a docked neck-linker configuration (C suffix) and the rest with an undocked (mobile) configuration (O suffix).

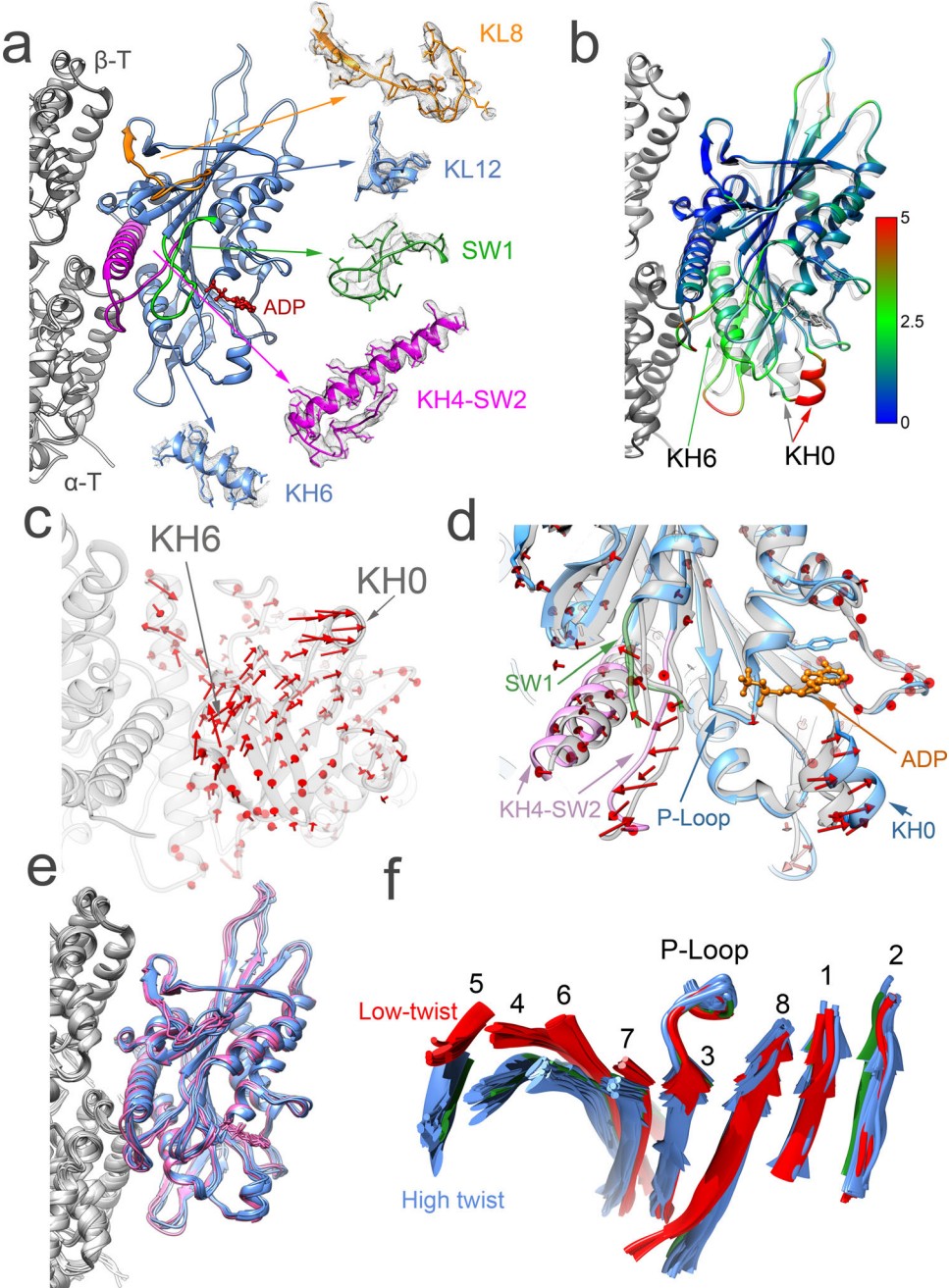

**Fig. 3 Microtubule-bound ADP and Apo structures. a** Side view of MT-K748-ADP atomic model structure. Insets show regions at the KIF14-microtubule interface (KL8, KL12, KH4, and KH6) and the switch loops (SW1 and SW2) with corresponding experimental density (gray mesh). **b** MT-K748-ADP (colored) and KIF14-ADP motor domain crystal structure (PDB: 4OZQ, semi-transparent gray) comparison. Structures are aligned to the KIF14 regions that interact with β-tubulin in the MT-K748-ADP structure (KL8 and KL12). The KIF14 motor domain of the MT-K748-ADP structure is colored by the distance to equivalent Cα atoms in the microtubule-unbound structure (semitransparent gray) according to the inset scale (in Å). **c** Displacement vectors (red arrows) between equivalent KIF14 motor domain Cα atoms in the MT-K748-ADP (light gray) and KIF14-ADP crystal structure. Structure alignment as in (**b**). **d** Nucleotide-binding pocket comparison between microtubule-unbound KIF14 (KIF14-ADP crystal structure, semitransparent gray) and microtubule-bound KIF14 (MT-K748-ADP, colored) structures. Both structures are aligned to their corresponding P-loops. **e** Superimposed MT-KIF14 complex structures (alignment as in **b**) in the ADP (pink) and Apo states (blue). **f** Superimposed central β-sheet of all the MT-KIF14 complex structures (aligned to their corresponding P-loops). Numbers corresponds to the kinesin motor domain β-strands KS1 to KS8. Based on the amount of twist of the central β-strand, the structures separate into two major groups: a more twisted group (blue and green) and a less twisted group (red). The low-twist structures (red) correspond to complexes in the closed conformation group and the high-twist structures to the semi-closed (green) and open/open* (blue) conformation groups. The orientation of the displayed structures relative to the microtubule is with the plus end up for (**a**), (**b**), and (**d**) and with the plus end away from the viewer in **c** and **f**. KH0, KH4, KH6: kinesin α-helix-0, 4, and 6. KL8, KL12 kinesin loop 8 and 12; SW1, SW2 kinesin switch-1 and switch-2 loops; α-T, β-T α- and β-tubulin.

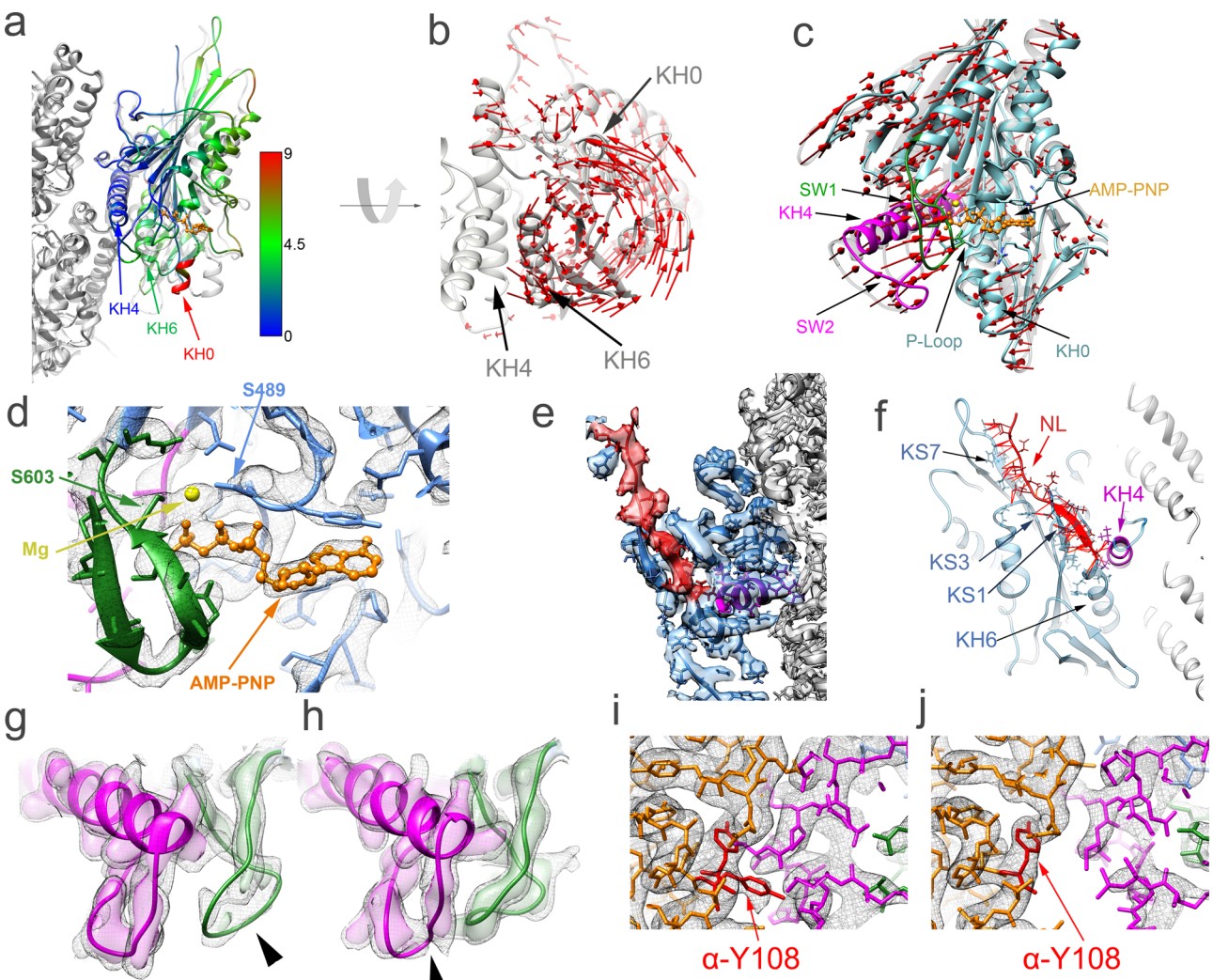

**Fig. 4 Closed nucleotide-binding pocket conformation. a** Superimposed MT-K748-ANP (colored) and MT-K748-ADP (semi-transparent gray) structures aligned to their corresponding tubulin β-chains. The KIF14 motor domain of the MT-K748-ANP structure is colored by the distance to equivalent Cα atoms in the MT-748-ADP structure according to the inset scale (in Å). **b** Displacement vectors (red arrows) between equivalent KIF14 motor domain Cα atoms in the MT-K748-ANP (light gray) and MT-K748-ADP structures after alignment as in (**a**). **c** Nucleotide-binding pocket comparison with displacement vectors (red) between MT-K748-ADP (semitransparent gray) and MT-K748-ANP (colored) structures after aligning their P-loops. **d** Detail of the nucleotide-binding pocket with corresponding densities (gray mesh) in the MT-K748-ANP structure. **e** Detail of the neck-linker domain with corresponding densities (semitransparent colored surface) in the MT-K748-ANP structure. **f** Neck-linker area in the MT-K748-ANP structure showing interacting residues with other areas of the motor domain. Pseudo-bonds between interacting atoms are represented as red lines (determined and displayed using the find clashes and contacts routine in UCSF-Chimera, see "Methods"). **g, h** Switch loop regions in the MT-K748-ADP (**g**) and MT-K748-ANP (**h**) structures. Corresponding densities are shown at two iso-density contour levels, higher (colored semitransparent surface) and lower (gray mesh). Densities that disappear at the higher contour level (presumably more disordered) are pointed with the black arrowheads. **i, j** Detail of the microtubule-KIF14 interface of the MT-K748-ADP (**i**) and MT-K748-ANP (**j**). Cryo-EM density shown as gray mesh iso-density surface. α-tubulin in orange except α-Y108, which is colored red. Note the distinct separation between α-tubulin and KIF14 residues and the alternate rotamer positions of the α-Y108 side chain in the open ADP and closed AMP-PNP structures. The KH4 and SW2 regions are colored magenta, the SW1 in green, the AMP-PNP in orange, the rest of the KIF14 motor in (**c-j**) in blue when present. Tubulin is colored solid gray in (**a**), (**e**), and (**f**). KS1, KS3, KS7 kinesin β-strand 1, 3 and 7; NL neck-linker. Other abbreviations as in Fig. 3 legend.

The MT-K735-AAF and MT-K743-AAF maps contain strong densities associated with ADP but densities that could have been attributed to AlF$_x$ or the Mg$^{2+}$ ion are near the noise level. The weak or absent AlF$_x$ densities and the lack of Mg$^{2+}$ coordination indicate a weakened interaction of the Pi mimic in the active site. This suggests that neck-linker undocking (e.g., under tension) and the consequent opening of the nucleotide-binding pocket promote Pi release from the active site after ATP hydrolysis.

**MT-K755 and MT-K772 one-head microtubule-bound structures.** The structure of the core motor domain of the MT-K755 and MT-K772 complexes in the Apo and ADP states conform to the open configuration, as it is also the case for the complexes with the shorter KIF14 constructs (Fig. 3e, Table 2, and Supplementary Table 1). However, different from the MT-K743-ADP/Apo and MT-K748-ADP/Apo maps, the MT-K755-ADP/Apo and the MT-K772-ADP/Apo maps show a neck-linker associated

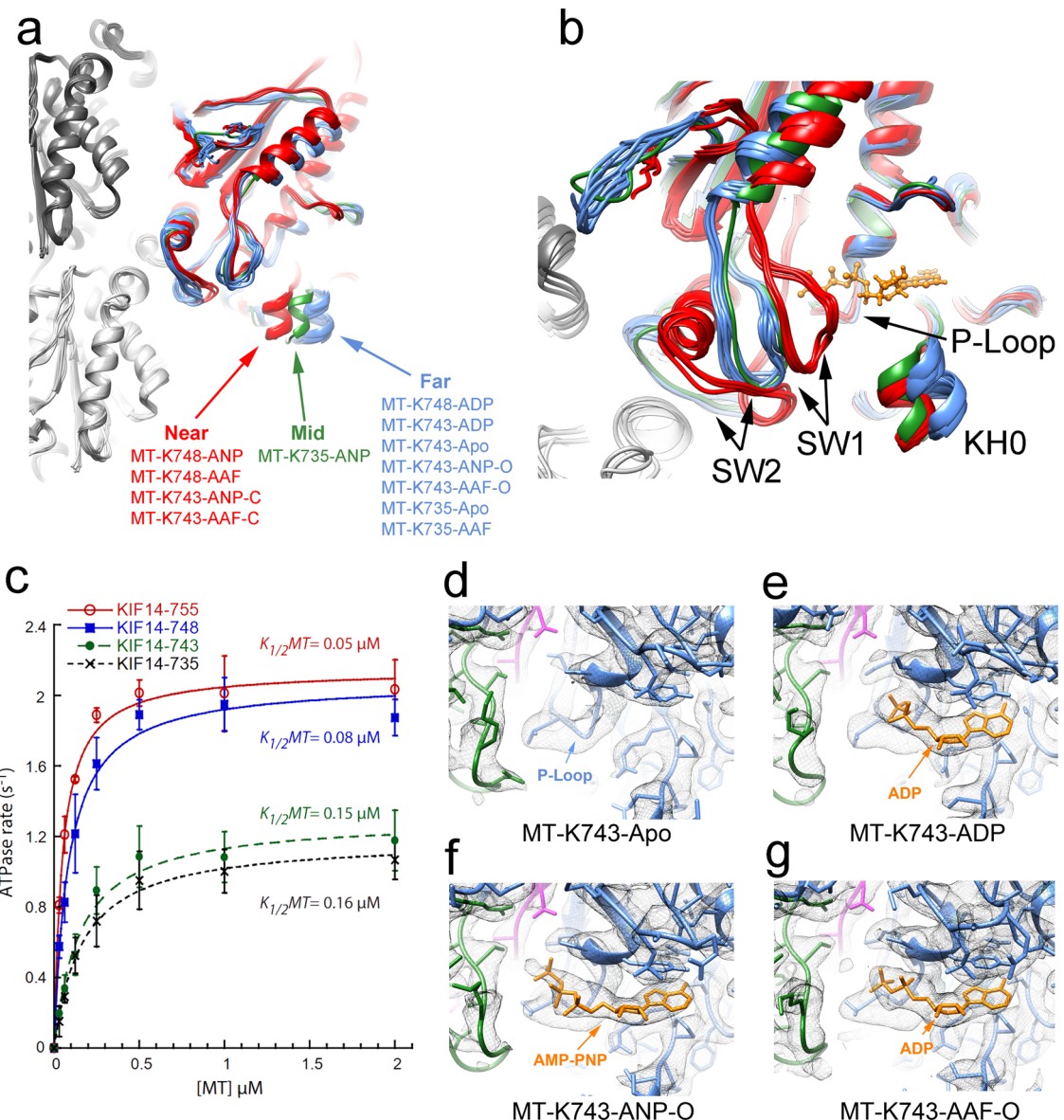

**Fig. 5 Neck-linker control of nucleotide-binding pocket closure. a** MT-KIF14 atomic models aligned to their corresponding β-tubulin chains. Three distinct locations of KH0, are observed: near, corresponding to the closed configuration (red); mid and far corresponding to the open/open* configurations (green and blue). **b** Nucleotide-binding pocket view showing the same structures in (**a**) with the same colors superimposed and aligned to their corresponding P-loops. **c** Microtubule-stimulated ATPase rate of K735, K743, K748, and K755 constructs. The basal ATPase rate for the four constructs was 0.3 s$^{-1}$ (subtracted from the total ATPase rate values). Error bars: standard deviation ($n = 3$). **d–g** Nucleotide-binding pocket detail of indicated structures with truncated neck-linkers all in the open configuration. Cryo-EM densities shown as an iso-density surface gray mesh, KIF14 model in blue with SW1 in green, SW2 in magenta, bound nucleotide (ADP or AMP-PNP) in orange.

density going sideways reaching the motor domain of an adjacent protofilament (Fig. 6a, b). Extra densities that could be associated with a partner motor domain in the dimeric K772 construct are not observed, suggesting that the partner head is not in a well-defined location. Another possibility to explain the lack of a distinct density associated with the partner motor domain would be that the two motor domains are bound to adjacent protofilaments in equivalent configurations. However, if this was the case, two alternative neck-linker orientations would be expected, one as observed and another going from helix-6 in the motor domain (where the neck-linker starts) to the dimerization domain at the tip of the observed neck-linker. Because such density is not observed, the map is better fitted with a model in which the KIF14 dimer in the Apo and ADP states binds to the microtubule in a one-head-bound state. In this state, one head is bound to the

microtubule with the nucleotide-binding pocket in the open configuration and the other is tethered and presumably in a semi-closed configuration similar to the microtubule-unbound KIF14-ADP crystal structure (K-ADP_4OZQ, Table 2).

**Coexisting motor domain conformations in KIF14 dimers.** Different from the Apo and ADP states in the AMP-PNP and ADP-AlF$_x$ states, the MT-K755 and MT-K772 helically averaged cryo-EM maps show a density connecting the motor domains along a protofilament (Fig. 6c–e). This density can be well-fitted with the undocked neck-linker of a motor domain positioned further toward the microtubule plus end (leading head) and part of the coiled coil domain connecting with a docked neck-linker of a motor domain positioned more toward the microtubule minus end

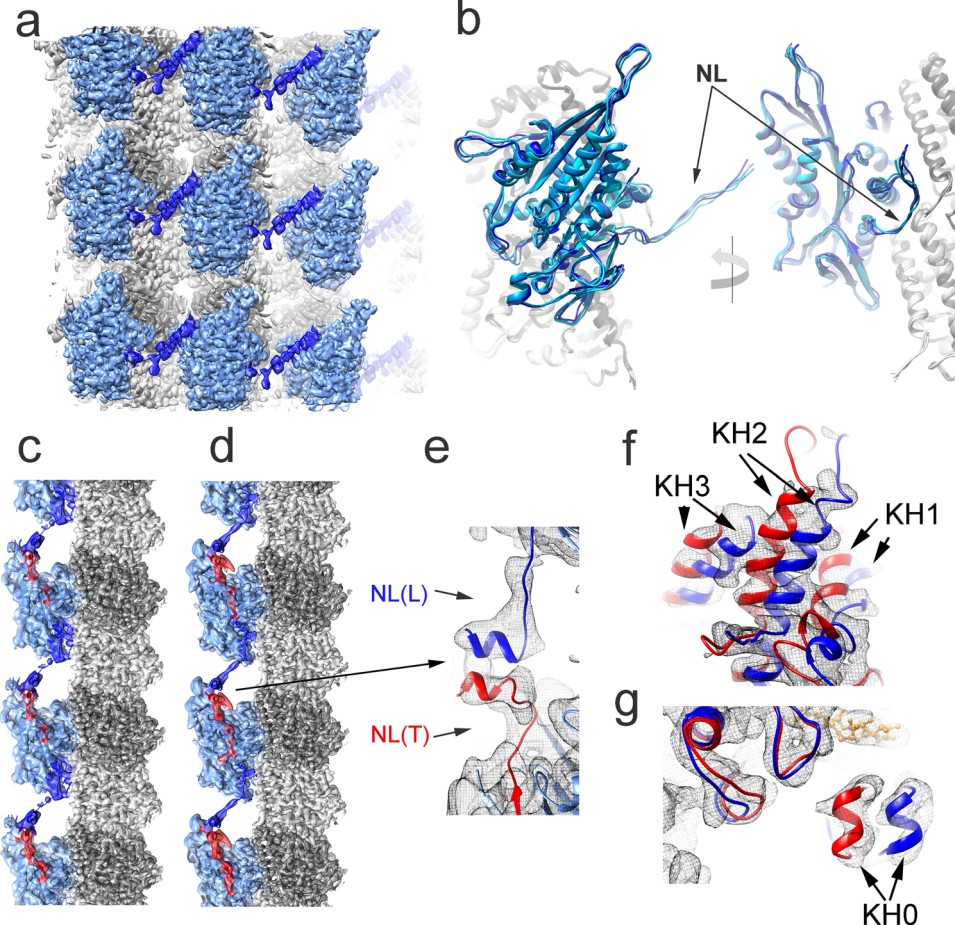

**Fig. 6 KIF14 dimer one- and two-heads microtubule-bound states. a** Iso-density surface representation of nine adjacent asymmetric units of the helically averaged MT-K755-ADP cryo-EM map colored according to the fitted atomic model regions. The KIF14 motor domain and neck-linker are colored light and dark blue, respectively. **b** Superimposed MT-K755-ADP, MT-K755-Apo, MT-K772-ADP, and MT-K772-Apo structures. All the structures aligned to their corresponding β-tubulin chains. The KIF14 motor domain of each model is colored in a different shade of blue. NL: neck-linker. **c** MT-K772-ANP helically averaged cryo-EM map. **d** MT-K755-ANP helically averaged cryo-EM map. In (**c**) and (**d**), the KIF14 motor domain is colored blue with docked and undocked neck-linker regions colored red and dark blue, respectively. **e** Detail of the connecting neck-linkers and part of the coiled coil region of trailing (red) and leading (blue) KIF14 motor domain structures fitted into the MT-K755-ANP helically averaged cryo-EM map (gray mesh). **f, g** MT-K755-ANP helically averaged cryo-EM map (gray mesh) with fitted open (blue) and closed (red) conformations of the KIF14 motor domain. **f** Shows the region of the motor domain with helices 1, 2, and 3 (KH1, KH2, KH3). **g** Shows the motor domain region near helix-0 (KH0). α- and β-tubulin regions colored light and dark gray, respectively, in (**a**–**d**).

(trailing head) (Fig. 6e). In addition, in these maps, the densities in the motor-domain area appears to be a combination of coexisting open and closed motor domain structures (Fig. 6f, g and Supplementary Fig. 10). The neck-linkers connecting density and the combined densities in the motor domain in the helically averaged cryo-EM maps strongly suggests that the KIF14 dimer in the AMP-PNP and ADP-AlF$_x$ states binds to the microtubule in a two-heads-bound configuration with leading and trailing heads in distinct conformations. To separate these conformations, we used the 3D refinement and classification procedure HASRC (next section).

**KIF14 microtubule two-bound-heads structures.** The structures of the leading and trailing heads of the MT-K755-ANP/AAF and MT-K772-ANP/AAF helically averaged cryo-EM maps were separated using the 3D refinement and classification procedure (HASRC, Supplementary Fig. 2). The procedure produced 3D class-averages containing two tubulin heterodimers, with two motor domains bound and connected by a density similar to the helically

averaged maps, but with clear differences between the structures of the leading and trailing motor domains (Figs. 2 and 7).

Similar two-bound-heads structures were observed with the K755 and K772 constructs (Fig. 2), even though only the K772 construct behaves as a dimer in solution (Fig. 1c). Construct K755 does not include the full CC1 domain but includes part of the first heptad repeat. Our results then show that this incomplete heptad, although insufficient to induce dimerization in solution, is able to do so when many motor domains are bound in close proximity in the microtubule lattice.

As inferred from the helically averaged map, the leading and trailing heads of the KIF14 two-heads-bound state are in two distinct conformations, open and closed (Fig. 7). The new maps with separated densities identify the leading head with the undocked backward pointing neck-linker as the one in the open conformation and the trailing head with a docked neck-linker as the one in the closed conformation. This is fully consistent with the structures of the monomeric KIF14 constructs indicating that AMP-PNP or ADP-AlF$_x$ induce the closed conformation only

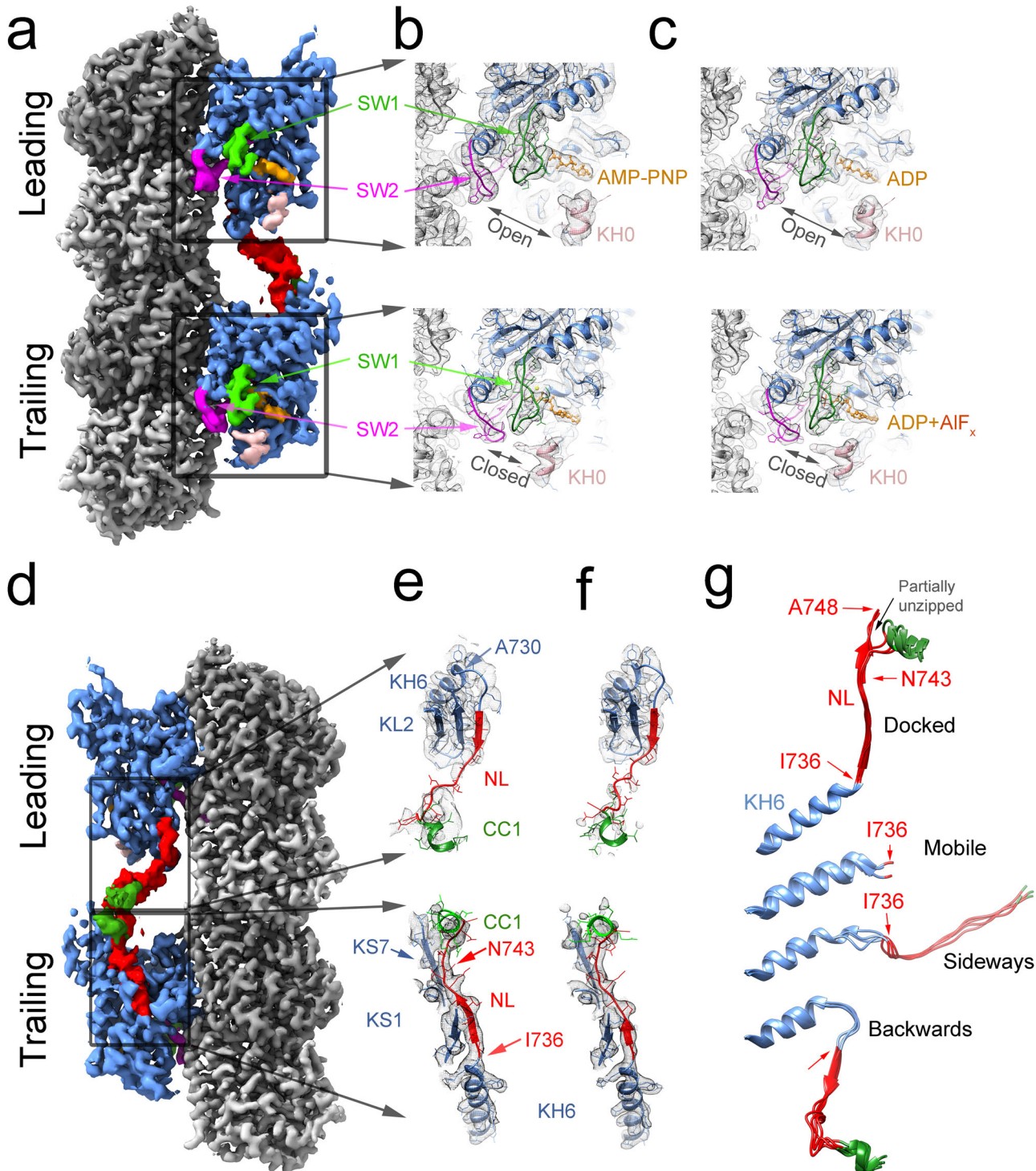

**Fig. 7 Leading and trailing KIF14 motor domain structures. a** Cryo-EM iso-density surface representation of the MT-K755-ANP HASRC refined two-heads-bound structure. **b** Detail of the nucleotide-binding pocket of the leading and trailing heads of the MT-K755-ANP structure. **c** Similar views as (**b**) of the MT-K772-AAF structure. **d** 180° rotation of structure shown in (**a**). **e** Detail of the KH6 and neck-linker regions of the leading and trailing heads of the MT-K755-ANP structure. **f** Similar views as (**e**) of the MT-K772-AAF structure. **g** Different KH6-neck-linker configurations observed in the MT-KIF14 structures (see Table 2 for their correspondence to particular structures). Cryo-EM densities shown as iso-densities colored surfaces (**a**) and (**d**) with the leading head neck-linker and coiled coil regions displayed at a lower contour level value than the rest of the map. α-tubulin in light gray, β-tubulin in dark gray. KIF14 motor domain in blue with SW1 in magenta, SW2 in bright green, KH0 in pink, the neck-linker in red, CC1 in dark green, and the bound nucleotide in orange. Same color scheme is used for the ribbon representation atomic models shown in (**b**, **c**, **e**–**g**). Cryo-EM map densities shown as gray mesh iso-density surfaces in (**b**, **c**, **e**, **f**).

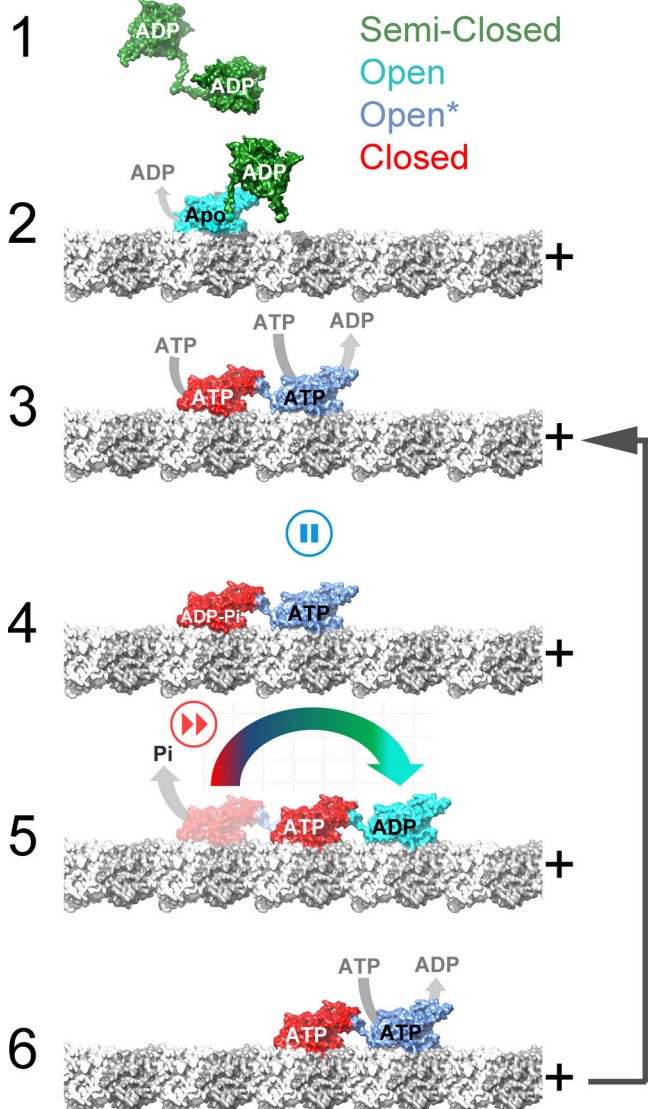

**Fig. 8 Kinesin dimer coordinated mechano-chemical cycle model.** The major KIF14 motor domain conformations determined, semi-closed, open, open\*, and closed (Table 2) are colored green, cyan blue, and red, respectively. α- and β-tubulin are colored light and dark gray, respectively, and the microtubule protofilaments are oriented with the plus end to the right. Starting from a microtubule-unbound semi-closed-ADP conformation (state 1) binding to the microtubule induces opening of the nucleotide-binding pocket and release of ADP (state 2). In this state, the neck-linker of both motor domains is undocked preventing the tethered motor domain from reaching the next tubulin binding site. The tethered motor domain remains mobile (disordered) and the dimer adopts a one-head-bound conformation similar to the MT-K755-Apo/ADP and MT-K772-Apo/ADP structures. Opening of the nucleotide pocket in the microtubule-bound head induces ADP release and allows ATP binding. ATP binding induces closure of the nucleotide-binding pocket and neck-linker docking. The docked neck-linker of the bound head positions the accompanying, tethered head, in a leading position (toward the microtubule plus end). The now leading head binds to the microtubule with its neck-linker undocked and pulled backward by the microtubule-bound trailing head. A two-head microtubule-bound intermediate similar to the MT-K755-ANP and MT-K772-ANP structures is formed (state 3). This two-heads-bound intermediate establish a coordinating gate in the chemo-mechanical cycle of the two motor domains. The undocked neck-linker in the leading head prevents it from adopting the closed catalytic conformation pausing hydrolysis until the trailing head releases from the microtubule. ATP hydrolysis is then favored in the neck-linker docked trailing head but not in the leading head leading to state 4. In this state, inter-head tension may cause partial undocking of the trailing head neck-linker accelerating product release and detachment of the trailing head leading to a one-head-bound intermediate (state 5). In the one-head-bound intermediate, the neck-linker of the ATP-bound leading head docks and places the tethered head forward toward the plus end. The two motor heads change leading and trailing positions and the now microtubule-bound leading head in the open configuration releases ADP and binds ATP (state 6). State 6 is similar to 3 but with the two heads exchanging leading and trailing positions and the whole molecule is displaced 8 nm in the plus direction.

closed structures, the neck-linker is docked and interacts with the core motor domain β-strands KS1 and KS7 (Fig. 4e, f). On the other hand, in the trailing head of the two-heads-bound structures, the neck-liker is partially unzipped as it separates from the core motor domain disrupting the interactions with KH7 (Fig. 7e–g). These interactions were shown to be necessary to stabilize the docked neck-linker configuration in the monomeric constructs (K748 vs. K743-ANP and AAF structures), also their disruption suggests that the docked neck-linker configuration of the trailing head is partially destabilized but held in place by the attached leading head. This could play a role during KIF14 translocation facilitating undocking of the neck-linker, opening of the nucleotide binding pocket, and releasing phosphate from the trailing head after ATP hydrolysis. The structural differences between the one-head- vs. the two-head-bound structures suggest that inter-head tension arises when the two-heads-bound intermediate forms.

## Discussion

We have solved the near-atomic resolution cryo-EM structures of five distinct KIF14 protein constructs bound to microtubules, in four nucleotide conditions mimicking key steps of the KIF14 ATPase cycle. Our data provides a comprehensive view at the highest resolution available of the mechano-chemical cycle of a motile kinesin and provide the structural basis for a mechano-chemical model for kinesin plus-end translocation (Fig. 8 and Supplementary Movie 2). In the following paragraphs, we further discuss the conclusions that can be drawn from the data

when the neck-linker is able to dock (Table 2). Also, consistent with the monomeric results, there are nucleotide-associated densities in both the leading and trailing heads, regardless of the motor domain being in the open or the closed conformation (Fig. 7a–c and Supplementary Fig. 11c–g). The MT-K755-ANP and MT-K772-ANP have AMP-PNP-associated densities in both the leading and trailing heads (Fig. 7a, b and Supplementary Fig. 11c, d, g). In the case of the MT-K755-AAF and MT-K772-AAF maps, there are also ADP-associated densities in both leading and trailing heads but AlF$_x$-associated densities only in the trailing head (Fig. 7c and Supplementary Fig. 11e, f, g). In the leading head, the situation is similar to the one observed in the monomeric MT-K735-AAF and MT-K743-AAF-O cryo-EM maps where densities that could have been attributed to the AlF$_x$ group are near the noise level.

Although the core motor domain structures of the two-heads-bound structures closely resemble the single-head-bound closed or open structures, there are structural differences in the neck-linker and KH6 helix. In the single-head-bound open structures, the neck-linker is either disordered (not resolved) or pointing sideways (Fig. 7g). On the other hand, in the leading head of the two-head-bound structure, the neck-linker is pulled backwards (toward the microtubule minus end) forming a β-sheet with KL2 and in addition KH6 partially unwinds (Fig. 7g). In the single-head-bound

and how they relate to previous studies and models of kinesin motility.

**Conformational changes induced by microtubule binding**. The microtubule-bound KIF14 open structures share some similarities with the near-atomic resolution crystal structure of kinesin-1 bound to a curved tubulin in the Apo state[18]. In both cases, the nucleotide-binding pocket becomes more open relative to the tubulin-unbound structure, but there are also significant structural differences between the complexes. In the kinesin-1-tubulin Apo structure, the switch-1 loop is mostly disordered and becomes ordered in the presence of ADP-AlF$_x$[16,18]. On the other hand, in all the KIF14-microtubule structures, including the Apo state, this loop is mostly ordered. The relative orientation of the motor domain relative to the bound tubulin is also slightly different (Supplementary Fig. 12). Differences are also noted when comparing the KIF14-microtubule complex structures and models derived from sub-nanometer resolution cryo-EM structures of the kinesin-1 microtubule complex[22]. How much of these differences are due to the different kinesin types (Kinesin-3 KIF14 vs. Kineins-1 KIF5B), different tubulin structures (curved vs. straight) or the resolution of the cryo-EM structures is not fully clear yet. To address this issue, structural studies of other kinesin-microtubule complexes at similar resolution are needed. Although there are several cryo-EM structures of kinesin microtubule complexes available, only recently it has become possible to achieve the resolution needed (<4 Å in the kinesin part of the map) to fully trace the polypeptide chain, to resolve the positions of side chains, to identify the nucleotide species in the active site, and to fully resolve the conformation of exposed and potentially mobile regions, such as the switch loops and the neck-linker.

**Microtubule and nucleotide-induced subdomain movements in the kinesin motor domain**. The transition between the major KIF14 motor domain conformations (semi-closed, open/open* and closed) can be approximated as movement between three subdomains we called the plus, minus, and switch subdomains (Supplementary Fig. 7). These subdomains overlap with previously defined subdomains based on the comparison of the structures of kineins-1 bound to tubulin in the Apo and ADP-AlF$_x$ states[18], but they are not identical. The transition between the open and closed conformations of the KIF14 motor domain can also be described as a clamshell-seesaw type movement between subdomains[22,36]. However, a distinction between our model and previous ones is that we recognize that the three moving subdomains form part of the tubulin–kinesin interface and the nucleotide-binding pocket is located between them. This, we propose is the key factor coupling the kinesin ATPase cycle with microtubule binding, as relative movements between the subdomains simultaneously alters the structures of the microtubule-binding interface and the nucleotide-binding pocket (Supplementary Figs. 7 and 8). Binding to the microtubule induces ADP release by producing a rotation of the minus subdomain relative to the plus subdomain that opens the nucleotide-binding pocket. Conversely, because closing of the nucleotide-binding pocket is also associated with a rotation of the minus subdomain, this process would be expected to be modulated by the interaction with the microtubule. Evidence for this can be found in the kinesin-13 family where an elongated loop-2 that extends the contacts between tubulin and the minus-subdomain prevents AMP-PNP-induced nucleotide-binding pocket closure when bound the microtubule (straight tubulin), but not when bound to the curved tubulin[37]. Modulation of the rotation between the plus and minus subdomains may also partly explain how mutations in KH6, which is part of the minus subdomain and the microtubule interface, reduce kinesin-1 microtubule gliding and microtubule-stimulated ATPase activity[38,39].

**The KIF14 neck-linker controls nucleotide-induced conformational changes**. There is good evidence indicating that binding of ATP or its analogs to the motor domain of microtubule-bound plus-end-directed motile kinesins, induce neck-linker docking[40–42] and closure of the nucleotide-binding pocket[20,22,43]. However, whether and how the neck-linker position could allosterically control the conformation of the motor domain and its catalytic activity was not clear. Our results show that without a docked neck-linker, the motor domain is prevented from adopting the closed catalytic conformation (Figs. 2, 5, Table 2, and Supplementary Table 1). This provides the structural basis for a fully reciprocal relationship between the neck-linker position and the nucleotide-binding pocket. Closure of the nucleotide-binding pocket induces neck-linker docking and conversely, preventing neck-linker docking impedes the nucleotide-binding pocket from adopting the closed catalytically active conformation. Our results also suggest that neck-linker undocking after ATP hydrolysis promote Pi release from the nucleotide-binding site.

It is plausible that the number of residues required to stabilize the docked conformation varies between kinesins as the neck-linker sequence is not highly conserved. This may provide a mechanism for fine tuning the catalytic activity of the motor, its response to tension, or the coordination between heads in a dimer. As shown here for KIF14, eight or less residues of the neck-linker (as in the K743 and shorter constructs) are insufficient to fully stabilize the docked conformation. In contrast, for the non-motile kinesin-13 KLP10A only four neck-linker residues are sufficient to stabilize a docked conformation and a closed nucleotide-binding pocket when the motor domain is bound to the curved tubulin[37].

The capacity of the kinesin motor domain to bind nucleotides, regardless of the position of the neck-linker or whether the nucleotide binding pocket is open or closed is unlikely to be a feature unique to KIF14. A recent structure of another kinesin (kinesin-13), where the nucleotide-binding pocket is prevented from closing by a different mechanism, also show an ATP analog bound in the active site[37]. Thus, controlling closure of the nucleotide-binding pocket and entering the hydrolysis competent step appears to be a general regulatory mechanism in the kinesin super-family. Although we cannot rule out small changes in nucleotide affinity, we do not expect them to be a limiting factor at physiological molar concentrations of ATP, similar to the ones used in this study.

**Leading and trailing motor domain structures of the KIF14 dimer**. The longer KIF14 constructs that includes part of the coiled coil dimerization domain, K755 and K772, show at near-atomic resolution what the structures of a kinesin dimer two-heads-bound intermediate looks like. Such intermediate is present in most models of kinesin plus-end-directed processive movement and it is likely to be the dominant state during translocation at physiological ATP concentrations[33,44,45]. The MT-K755-ANP/AAF and MT-K772-ANP/AAF complexes show the leading and trailing heads in two distinct conformations providing structural evidence of coordination between the two motor domains. The ATPase cycles of the two motor domains are kept out of phase as the undocked neck-linker of the leading head keeps the catalytic site in the open configuration inhibiting ATP hydrolysis (Fig. 8 steps 3 to 4). This control mechanism where the ATPase cycle of the leading head is slowed down by the backward pulling rear

head has been referred as front-head gating[32] and our data provides direct structural evidence for it. However, distinct from previously proposed models, our data argue that the step of the ATPase cycle that is inhibited in the leading head is ATP hydrolysis, rather than nucleotide binding, as has been proposed based on single molecule force mechanics and kinetic experiments[46–48]. If nucleotide binding was prevented from binding the leading kinesin head when the neck-linker is forced out of the docked configuration, then an empty nucleotide-binding pocket would be expected in the leading head or in the monomeric KIF14 constructs with truncated neck-linkers (K735 and K743). Instead, the nucleotide in solution is always found bound, regardless whether it is the leading or the trailing head or whether the active site is open or closed (Fig. 5e–g and Supplementary Fig. 11b–g). This is unlikely to be a KIF14-specific feature as a previous medium resolution cryo-EM structure of a kinesin-1 two-heads-bound intermediate also suggested that nucleotides were present in both leading and the trailing heads[49]. The structure of the two-head intermediate also reveals another potential point of regulation. The partially unzipped neck-linker in the trailing head could in principle accelerate opening of the nucleotide-binding pocket and product release leading to rear head detachment (step 4 to 5).

The dimeric K772 construct in the Apo and ADP states also provide a high-resolution view of the one-head microtubule-bound intermediate occurring during kinesins translocation when one kinesin motor domain changes binding sites along the microtubule. In this intermediate, the bound motor domain is in the open configuration with an undocked neck-linker pointing to the side and the partner motor head is mobile and not visible in the cryo-EM maps (Fig. 6a, b). This fits well with data indicating that kinesin-1 dimers during translocation form a transient one-head microtubule-bound intermediate with a highly mobile tethered head in the so-called ATP-waiting state[44,45,50]. The conformation of the neck-linker in the K772 and K755 Apo/ADP structures suggests that the tethered head is positioned sideways and toward the microtubule plus end (the direction of movement) relative to the trailing position seen in the two-head microtubule-bound intermediate (as depicted in Fig. 8, step 2).

In summary, we present near-atomic resolution structures of twenty distinct kinesin KIF14 microtubule complexes providing the structural basis for a coordinated translocation mechanism where the hydrolysis step of each motor domain is controlled by the conformations of their connecting neck-linkers domains (Fig. 8). Given the high degree of motor domain homology, and that KIF14 shares a similar quaternary structure with other plus-end-directed motile kinesins, most aspects of the proposed model are likely to be generally applicable to this group of motors. In addition, we expect that the image analysis procedures introduced to separate coexisting KIF14 conformations would be useful in future structural studies of other kinesin and microtubule-binding protein complexes.

## Methods

**Protein purification**. KIF14 constructs were generated as GST-fusion proteins by amplification of the desired constructs from a plasmid containing the corresponding cDNA using polymerase chain reactions. The plasmids were transformed into BL21 (DE3) pLysS cells and were induced with 0.5-mM IPTG at 18 °C overnight. GST-fusion proteins were purified on glutathione resins (Genscript)[14]. KIF14 constructs were cleaved on beads with PreScission protease in BRB80-based cleavage buffer (80-mM PIPES pH 6.8, 1-mM EGTA, 1-mM MgCl₂, 100-mM KCl, 1-mM DTT, and 0.1-mM ATP) to generate the untagged motors. KIF14 constructs were flash-frozen in small aliquots in liquid nitrogen, and stored at −80 °C freezer.

Fifteen-protofilaments-enriched microtubules were prepared from porcine brain tubulin (Cytoskeleton, Inc. CO)[37].

**ATPase activity assay**. A malachite green-based phosphate detection assay was used to measure KIF14-mediated ATPase activity[14,51]. Briefly, reactions were

assembled in BRB40-based buffer (40-mM PIPES pH 6.8, 1-mM MgCl₂, 1-mM EGTA, 20-μM paclitaxel (Taxol®), 25-mM KCl, 0.25-mg/mL BSA, 1-mM DTT, 0.02% Tween), supplemented with 1-mM ATP, 2 μM of paclitaxel-stabilized microtubules, and 50-nM KIF14 protein constructs. Basal activity of the KIF14 constructs was determined using the same reaction condition with no microtubules added. Reactions were allowed to proceed for 10 min, quenched with perchloric acid and malachite green reagent[14]. The signal was quantified by the absorbance at 620 nm in a Genios Plus plate reader (Tecan).

**Cryo-EM KIF14-MT complexes sample preparation**. MT-K735-ANP, MT-K735-Apo, and MT-K772-ANP datasets were collected on untreated carbon/copper grids (Quantifoil R2/2 300 mesh). The grids used for the other 15 datasets were gold grids (UltrAuFoil R2/2 200 mesh) plasma cleaned just before use (Gatan Solarus plasma cleaner, at 15 W for 6 s in a 75% argon/25% oxygen atmosphere). Microtubule was polymerized fresh the day of the cryo-EM sample preparation. Kinesin aliquots were thawed on ice just before use. All nucleotides stock solutions used were prepared with an equimolar equivalent of MgCl₂ to ADP or AMP-PNP. Four microliters of a microtubule solution at 2–5-μM tubulin in BRB80 plus 20-μM paclitaxel were layered onto the EM grid and incubated 1 min at room temperature. This microtubule solution also contains either AMP-PNP at 4 mM, ADP at 4 mM, or ADP at 4 mM plus 2-mM AlCl₃ and 10-mM KF. During the incubation time, a fraction of the thawed kinesin aliquot was diluted to prepare a 20-μM kinesin solution (except 10 μM for K772 in all nucleotide conditions and K748-ADP) containing 20-μM paclitaxel and either of the four nucleotide conditions to be probed: (1) 4-mM ADP, (2) apyrase: $5 \times 10^{-3}$ units per μl (Apo conditions), (3) 4-mM AMP-PNP, and (4) 4-mM ADP plus 2-mM AlCl₃, and 10-mM KF (ADP-AlF$_x$ conditions). Then the excess microtubule solution was removed from the grid using a Whatman #1 paper. Four microliters of the kinesin solution was then applied onto the EM grid, transferred to the chamber of a Vitrobot apparatus (FEI-ThermoFisher MA) at 100% humidity where it was incubated for 1 min at room temperature, and blotted for 2.5–3 s with a Whatman #1 and −2-mm offset before plunge-freezing into liquid ethane. Grids were clipped and stored in liquid nitrogen until imaging in a cryo-electron microscope.

**Cryo-EM data collection**. Data were collected at 300 kV on Titan Krios microscopes equipped with K2 summit detectors (Supplementary Table 1). Acquisition was controlled using Leginon[52] with the image-shift protocol and partial correction for coma induced by beam tilt[53]. Data collection was mainly performed semi-automatically using 3–5 exposures per 2-μm diameter holes. The exposures were fractionated on 35–50 movie frames. The defocus ranges and cumulated dose are given in Table 1.

**Helical-single-particle 3D reconstruction**. Movie frames were aligned with motioncor2 v1.0 or v1.1 generating both dose-weighted and non-dose-weighted sums. All the datasets collected with a pixel size below 1 Å were corrected for magnification anisotropy (Supplementary Table 2). Before each of the corresponding session, a series of ~20 micrographs on a cross-grating calibration grid with gold crystals was used to estimate the current magnification anisotropy of the microscope using the program mag_distortion_estimate v1.0.1[54]. Magnification anisotropy correction was performed within motioncor2 using the obtained distortion estimates (Supplementary Table 2). Contrast transfer function (CTF) parameters per micrographs were estimated with Gctf (v1.06)[55] on aligned and non-dose-weighted movie averages.

Images of 15R microtubules were processed using a helical-single-particle 3D analysis workflow[37] to produce two independently Frealign[56] refined maps from the two datasets halves, limiting the refinement data to (1/8) Å⁻¹ and run until no further improvement in resolution was detected. Number of particle images and asymmetric units included in the 3D reconstructions are reported in Table 1 and the particle-boxes sizes in Supplementary Table 2. A cleaning step was performed for several datasets (Supplementary Table 2): particles for which the Euler angles are not compatible with 15R symmetry (either because not 15R or because they may be 15R particles poorly aligned) were discarded (<4% of the data). If the resolution at that stage was at ~3.6 Å or better, per-particle image CTF values were refined. The datasets that underwent a CTF refinement step are listed in Supplementary Table 2. For such datasets, the defocus of each particles in each half dataset were refined using one cycle of FrealignX[57] against their respective half-reconstruction and without particle alignment (defocus search parameters: step: 50 Å, range: 1250 Å). A moving median over five contiguous particles was applied on each microtubule to assign their final per-particle defocus values. After CTF refinement, further Frealign cycles limiting the refinement data to (1/5) Å⁻¹ were run until no further improvement in resolution was detected. A Frealign 3D classification into two classes was used to improve the signal to noise (S/N) of the 748-ANP dataset. Unmasked–unfiltered–unsharpened Frealign reconstructions were obtained for each half dataset and helical symmetry was imposed in real space to each half map with relion_helical_toolbox. These half maps were merged and corrected for the modulation transfer function of the detector by relion_postprocess.

To obtain a final locally filtered and locally sharpened map post-processing of the pair of unfiltered and unsharpened half maps was performed as follows. One of the two unfiltered half-map was low-pass-filtered to 15 Å and the minimal

threshold value that does not show noise around the helical segment was used to generate a mask with relion mask create on 85% of the helical segment on its helical axis (low-pass filtration: 15 Å, extension: 10 pixels, soft edge: 10 pixels). This soft mask was used in blocres[58] on 12-pixel size boxes to obtain initial local resolution estimates. The merged map was locally filtered by blocfilt[58] using blocres local resolution estimates and then used for local low-pass filtration and local sharpening in localdeblur[59] with resolution search up to 25 Å. The localdeblur program converged to a filtration appropriate for the tubulin part of the map but over-sharpened for the kinesin part. The maps at every localdeblur cycle were saved and the maps with optimal filtration for the kinesin part area were selected. Finally, helical symmetry was imposed to the locally filtered maps.

**Helical assembly subunit refinement and classification (HASRC).** The helical-single-particle 3D reconstruction procedure generates density maps where all the asymmetric units in the helical assembly (one tubulin heterodimer with one-bound kinesin motor domain) are averaged. To independently determine the structure of the two motor domains of kinesin dimers bound to the microtubule (two-heads-bound), we implemented a procedure (HASRC) to isolate and classify individual subunits in a helical assembly. Different from a previous method[49], HASRC can separate two-bound kinesin dimer complex structures in fully decorated microtubules at near-atomic resolution. HASRC was implemented in Relion 3.1 adapting ideas and methods previously used to separate and classify the subunits contained in symmetrical multi-subunit assemblies[60,61]. These ideas have also been used recently to refine the structure of microtubule protofilaments[62]. We implemented specific subunit location refinement and classification steps in HASRC that were essential to separate coexisting kinesin motor domain conformations at high resolution.

In addition to the ability to separate coexisting subunit structures, HASRC allowed to account for local lattice distortions, which resulted in up to 0.4-Å resolution improvement over the helically averaged maps. Thus, to produce maps at the highest possible resolution, we applied HASRC to all datasets, dimers in the two-heads-bound states as well as monomers and dimers in the one-head-bound states.

For each of the four KIF14 dimer two-heads-bound states datasets (K755 and K772, ANP and AAF states), the following procedure was used (Supplementary Fig. 2):

(1) Relion helical refinement. The two independent Frealign helical refined half datasets were subjected to a single helical autorefinement in Relion 3.1 where each dataset was assigned to a distinct half-set and using as priors the Euler angle values determined in the helical-single-particle 3D reconstruction (initial resolution: 8 Å, sigma on Euler angles sigma_ang: 1.5, no helical parameter search).

(2) Asymmetric refinement with partial signal subtraction. An atomic model of a KIF14 dimer two-heads-bound state was used to generate two soft masks (Supplementary Fig. 2b, c) using EMAN pdb2mrc and relion_mask_create (low-pass filtration: 30 Å, initial threshold: 0.05, extension: 14 pixels, soft edge: 6 or 8 pixels). One mask (mask$_{full}$) was generated from a KIF14 dimer model bound to two tubulin dimers while the other mask (mask$_{kinesin}$) was generated with only the kinesin coordinates. The helical dataset alignment file was symmetry expanded using the 15R microtubule symmetry of the dataset. Partial signal subtraction was then performed using mask$_{full}$ to retain the signal within that mask. During this procedure, images were re-centered on the projections of 3D coordinates of the center of mass of mask$_{full}$ ($C_M$) using a box size as indicated in Supplementary Table 2. The partially signal subtracted dataset was then used in a Relion 3D autorefinement procedure using as priors the Euler angle values determined form the Relion helical refinement and the symmetry expansion procedure (initial resolution: 8 Å, sigma_ang: 2, offset range corresponding to 3.5 Å, healpix_order and auto_local_healpix_order set to 5). The CTF of each particle was corrected to take into account their different position along the optical axis.

(3) 3D classification of the kinesin signal. A second partial signal subtraction procedure identical to the first one but using mask$_{kinesin}$ and with particles re-centered on the projections of $C_M$ was performed to subtract all but each pair of kinesin signals (Supplementary Fig. 2). The images obtained were resampled to 3.5 Å/pixel and the 3D refinement from step 2 was used to update the Euler angles and shifts of all particles. A 3D focused classification without images alignment and using a mask for the kinesin generated like mask$_{kinesin}$ was then performed on the resampled dataset to separate the kinesin states (8 classes, tau2_fudge: 4, padding: 2, iterations: 175). Two of the resulting classes contained two well-resolved kinesin motor domains while the others had absent or not well-resolved kinesin densities (Supplementary Fig. 2f). The two classes with well-resolved kinesin densities differed in the location of the density connecting the two kinesin motor domains: one class with the connecting density at the center corresponding to a dimer with leading and trailing kinesin heads and the other class corresponding to the two unconnected kinesin heads of two distinct dimers. These two classes were equally populated as expected from the procedure used, which samples the microtubule axially at each tubulin heterodimer, rather than the two heterodimers span of the kinesin dimer. For the much smaller MT-K772-ANP dataset, another classification strategy had to be

used: first a focused classification (3 classes, tau2_fudge: 6, padding: 2, iterations: 25) was used to eliminate the particles generating a low-resolution class average and an undecorated class average. A second focused classification (2 classes, tau2_fudge: 16, padding: 2, iterations: 175) enabled to separate the main class of the first classification into two similarly populated dimer configurations.

(4) Subunit refinement. The subset of particles belonging to the class with a centered isolated dimer was further refined using a Relion 3D autorefinement with the same parameters used in step 2.

(5) 3D reconstructions with original images (not signal subtracted). To avoid potential artefacts introduced by the signal_subtraction procedure, final 3D reconstructions were obtained using relion_reconstruct on the original image-particles without signal subtraction. Map filtration was then performed the same way as with the helically averaged maps, without symmetry imposition.

For the one-head-bound datasets (K735, K743, and K748 in all nucleotide states and K755 and K772 in the Apo and ADP states) the same procedure as described above was employed with the following modifications (Supplementary Fig. 3): in step 2, the mask$_{full}$ was generated with a PDB containing 1 kinesin motor bound to 1 tubulin dimer and two longitudinally flanking tubulin subunits. The mask mask$_{kinesin}$ was generated with a KIF14 motor domain model. All the datasets produced at least one class where the kinesin motor densities were well-resolved. In all cases, only a single motor domain configuration was found except for the K743 construct in the ANP and AAF states where two well-resolved classes with two different kinesin conformations (open and closed) were found (Supplementary Fig. 3h–k). No further refinement was performed after classification (no step 4). The presence of two coexisting conformation in these datasets is consistent with the mixture of open and closed configuration densities observed in the helically averaged maps (Supplementary Fig. 10).

**Cryo-EM resolution estimation.** The final resolutions for each cryo-EM reconstruction were estimated from FSC curves generated with Relion 3.1 postprocess (FSC$_{0.143}$ criteria, Table 1 and Supplementary Fig. 1). To estimate the overall resolution, these curves were computed from the two independently refined half maps (gold standard) using soft masks that isolate a single asymmetric unit containing a kinesin and a tubulin dimer. The soft masks were created with Relion 3.1 relion_mask_create (low pass filtration: 15 Å, threshold: 0.1, extension: 2 pixels, soft edge: 6 pixels) applied on the correctly positioned Eman1 pdb2mrc density map generated with the coordinates of the respective refined atomic models. FSC curves for the tubulin or kinesin parts of the maps were generated similarly using the corresponding subset of the PDB model to mask only a kinesin or a tubulin dimer (Table 1).

The final HASRC refined cryo-EM maps together with the corresponding helically averaged maps, half maps, masks used for resolution estimation and partial signal subtraction and the FSC curves are deposited in the Electron Microscopy Data Bank (Table 1). For the MT-K755 and MT-K772 cryo-EM datasets, composite maps were made from the localdeblur maps where the noisier regions (near the coiled coil domain) were low-pass-filtered (7–8 Å) and rescaled. For each of these datasets, the original localdeblur map (no-composite) and the low-pass-filtered map are deposited as additional maps together with the mask used to make the composite map.

**Model building.** Atomic models of the cryo-EM density maps were built as follow. First, atomic models for each protein chains were generated from their amino-acid sequence by homology modeling using Modeller[63]. Second, the protein chains were manually placed into the cryo-EM densities and fitted as rigid bodies using UCSF-Chimera[64]. Third, the models were flexibly fitted into the density maps using Rosetta for cryo-EM relax protocols[65,66] and the models with the best scores (best match to the cryo-EM density and best molprobity scores) were selected. Fourth, the Rosetta-refined models were further refined against the cryo-EM density maps using Phenix real space refinement tools[67]. Fifth, the Phenix-refined models were edited using Coot[68]. Several iterations of Phenix real space refinement and Coot editing were performed to reach the final atomic models.

Atomic models and cryo-EM map figures were prepared with UCSF-Chimera[64] or VMD[69]. Movies were made with VMD[69] and R[70].

**Model coordinates precision.** To estimate the precision of the model atomic coordinates fitted into the cryo-EM density maps, we used the method proposed by Herzik et al.[71] that test the convergence of independently fitted atomic model to the cryo-EM density maps. Each coordinate model was refined against the corresponding cryo-EM density map using Rosetta[65,66]. Two hundred independent models were generated, 100 using the refine-torsion protocol and 100 using the refine-cartesian protocol. The ten best models produced by each protocol (best fit to density and best molprobity scores) were then subjected to real-space refinement in Phenix[67]. Precision was expressed as the root mean square deviation between equivalent Cα carbon coordinates of the 20 resulting models (Supplementary Fig. 5).

**Cryo-EM density quantification.** The relative intensity of different regions of the cryo-EM maps (Supplementary Figs. 10e and 11g) were determined from density

maps and the corresponding fitted atomic models using the measure mapValues command of UCSF-Chimera[64]. When comparing the density of equivalent regions in different maps and atomic models, the maps and models were first aligned either to their corresponding β-tubulin (open vs. closed densities, Supplementary Fig. 10e) or to the P-loop (nucleotide relative density, Supplementary Fig. 11g) using the UCSF-Chimera matchmaker and matrixcopy commands. Densities were calculated as the average density near the atoms of the ligands or the backbone of specific KIF14 residues in the atomic models. Background density was estimated with a set of atoms placed near the selected residues but outside the area occupied by the atomic models. The nonoverlapping open and closed conformation regions were defined as KIF14 residues 403–412, 463–471, 501–515, and 577–579 (corresponding to KH0, KH1, KH2, and KH3 regions) in the MT-K748-ADP and MT-K748-ANP models, respectively. Relative density for each map was calculated as the average density at each position minus background (open or closed) over the total (open + closed). Nucleotide relative density was calculated as the average density of the nucleotide base atoms (ADP or AMP-PNP) over the average density at residues of the P-loop (KIF14 residues 476–494) in the same map.

**Reporting summary**. Further information on research design is available in the Nature Research Reporting Summary linked to this article.

## Data availability

Data supporting the findings of this manuscript are available from the corresponding authors upon reasonable request. A reporting summary for this Article is available as a Supplementary Information file. Atomic coordinates and corresponding cryo-EM density maps (HASRC refined maps and helically averaged maps), including the half maps, masks and FSC curves used to estimate spatial resolution have been deposited in the Protein Data Bank (PDB) and Electron Microscopy Data Resource (EMD) under the accession codes 6WWE, 6WWF, 6WWG, 6WWH, 6WWI, 6WWJ, 6WWK, 6WWL, 6WWM, 6WWN, 6WWO, 6WWP, 6WWQ, 6WWR, 6WWS, 6WWT, 6WWU, 6WWV, 7LVQ, 7LVR and EMD-21932, EMD-21933, EMD-21934, EMD-21935, EMD-21936, EMD-21937, EMD-21938, EMD-21939, EMD-21940, EMD-21941, EMD-21942, EMD-21943, EMD-21944, EMD-21945, EMD-21946, EMD-21947, EMD-21948, EMD-21949, EMD-23540, EMD-23541 (Table 1). Source data are provided with this paper.

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

## Acknowledgements

This work was supported by NIH Grant R01GM113164 (H.S.), Canadian Cancer Society Research Institute Grant #703405 (B.H.K.), and Natural Sciences and Engineering Research Council of Canada Discovery Grant RGPIN/04103-2015 (B.H. K.). B.H.K. is a recipient of the Fonds de Recherche du Québec–Santé (FRSQ) Chercheure-Boursière Junior 1, Junior 2 Awards, and the Canadian Institutes of Health Research (CIHR) New Investigator Award. Cryo-EM data collection was performed at the Simons Electron Microscopy Center and National Resource for Automated Molecular Microscopy located at the New York Structural Biology Center, supported by grants from the Simons Foundation (SF349247), NYSTAR, and the NIH National Institute of General Medical Sciences (GM103310) with additional support from Agouron Institute (F00316) and NIH (OD019994). We thank Laura Yen, Misha Kopylov, Daija Bobe, Ed Eng, and Bill Rice for assistance and support during data collection. We thank The Albert Einstein College of Medicine (AECOM) Analytical Imaging Facility for electron microscopy support and the AECOM High Performance Computing Facility for computing support.

## Author contributions

M.P.M.H.B. designed experiments, collected data, performed 3D reconstructions and digital image data analysis including the development of procedures to separate and classify coexisting conformations in helical assemblies, A.B.A. made cryo-electron microscopy samples and collected data. M.P. and S.D. produced protein constructs and performed biochemical assays. B.H.K. performed biochemical assays and designed protein constructs. H.S. supervised the project, analyzed data, and wrote first draft of the manuscript. All authors discussed the results and edited the final manuscript version.

## Competing interests

The authors declare no competing interests.
