## [Peer Review File · Nature Communications]

Reviewer #1 (Remarks to the Author):

This excellent work uses cryo-EM to provide the highest resolution and most comprehensive structural description of a kinesin motor to date. By looking at several different constructs, both monomeric and dimeric, the authors fill in important details of the motility cycle that were previously not well established. An especially noteworthy contribution is the discovery that binding of ATP (modeled by nonhydrolyzable analog AMPPNP) by itself is not sufficient to allosterically transform kinesin to its 'closed', actively hydrolyzing conformation. The experiments here demonstrate this by showing that if the neck linker is truncated, ATP analog binding fails to fully close the nucleotide pocket. Supporting biochemical data are consistent with the structures, showing a somewhat reduced ATPase activity with the truncation.

Overall, the manuscript is clearly written, well organized and the figures are beautifully rendered. I have no major criticisms. I attach a few minor suggestions, but I feel the manuscript could be published essentially as is.

P. 5. 'On the other hand, the microtubule bound cryo-EM structures show clear densities along the full length of these regions '.

-> Ordering of SWI/SWII upon MT or tubulin binding has been observed before in cryo and X-ray, and it would not hurt to cite the previous works that found this (Atherton...Moore, 2013; Gigant...Knossow, 2014; Shang... Sindelar, 2014;).

P. 7. 'The motor domain in the MT-K735-ANP complex shows a new configuration where the KH0-containing-sub-domain is rotated to an intermediate position between the open and closed configurations '.

-> It's not clear to me that this is indeed a new conformation rather than a sum of fractionally occupied open and closed states. The authors could consider referencing Sindelar and Downing, 2010 where evidence was presented that AMPPNP bound kinesin-MT monomers are in equilibrium between open and closed states. The main evidence that MT-K735-ANP is unique conformation seems to come from the density for helix-0 in Supp. Fig3 (green) but I think this kind of density could easily arise from a superposition - resolution is not high enough to say for sure.

P. 10. 'Consistent with the structural results, we found that the K735 and K743 constructs have reduced microtubule stimulated ATPase activity relative to the longer KIF14 constructs'.

-> I encourage the authors to reference and cite Gigant...Knossow, 2014, here, as they found similar (actually, much greater) effects in a truncated neck linker mutant using ATPase assays.

MINOR EDITS:

P. 5: observed with the *construct* that include include at least 15 neck linker residues

-> constructs

P. 12; even though only the K772 *constructs*

-> construct

p. 15: The fact neck-linker docking -> "fact that"

Reviewer #2 (Remarks to the Author):

This paper by Benoit et al describes the cryo-EM structures of mitotic kinesin, KIF14. They used five constructs with different length (they all have full-length motor domain and have different length of neck linker or dimerization coiled-coil region). Authors took EM data sets in four different nucleotide-state (ADP, apo, AMPPNP, and ADPAIFx) for each construct. In total, they solved 18 unique EM maps at 2.9 to 4 Å resolution and made 22 unique atomic models. The cryo-EM analysis is technically sound and quite extensive.

Using these structures, the authors discuss about the ATP-dependent structural changes in KIF 14 i.e.

- (1) Structural changes of KIF14 induced by microtubule binding.
- (2) Structural changes after ATP binding and followed by its hydrolysis, (3) Allosteric regulatory mechanisms for the structural changes involved in hydrolysis by neck linker docking.
- (4) The mechanochemical cycle of dimer KIF14 based on their dimer atomic models.

The most important point in this paper is that they have determined the structure of the motor domain-microtubule complex with various lengths of constructs in various nucleotide states, and they classified them into 4 group (semi-closed, open, open* and closed), allowing them to discuss in detail about the nucleotide-dependent conformational changes in KIF14. It is particularly valuable that their new ATP or ADP Pi analogue structure from truncated neck linker constructs allow them to discuss about the allosteric regulatory mechanisms for the structural changes involved in hydrolysis by neck linker docking.

The reviewer think that their experimental results and the discussion about the monomeric KIF14 motor domain are of general interests to biophysicists studying biological motors.

On the other hand, their structural model of the dimer are not consistent with the current understanding of conventional kinesin movements. Therefore, it require additional experimental data. Therefore, the reviewer suggests revising the interpretations and models of dimer KIF14.

Major points:

The authors created dimer models and used it as the basis of Movie 2. However, it require more experimental data to support the model shown in the movie.

(1) If the authors want to support the model based on structural studies, they have to distinguish the leading and trailing head of KIF14 as Liu et al (2017) did for conventional kinesin. In the current manuscript, the 3D reconstruction was done assuming that there are no differences in the leading and trailing heads. Therefore, the current 3D reconstructions of K772 are the average of the leading and trailing heads. The authors derived two possible models of the dimer structure from a single averaged structure map.

(2) The authors do not consider the possibility of artifacts due to the excess amount of nucleotides to the solution.

In fact, the authors present a structure in which ADP-AIFx (ADP Pi analog) is attached to both heads and both heads take open conformations in movie2. Such structures are achieved only in mutants with extended neck linker, but not in the wild-type for conventional kinesin, as demonstrated by single-molecule measurements (Andreasson et al 2015) from Block lab. It should be shown whether the dimer models they present in this paper is possible in the case of KIF14.

References

Structural basis of cooperativity in kinesin revealed by 3D reconstruction of a two-head-bound state on microtubules.

Liu, D., Liu, X., Shang, Z., Sindelar, C. V.

Elife 2017 6:

Examining kinesin processivity within a general gating framework.

Andreasson, J. O., Milic, B., Chen, G. Y., Guydosh, N. R., Hancock, W. O., Block, S. M.

Elife 2015 4:

Reviewer #3 (Remarks to the Author):

Benoit et al present a cryo-EM study of the interaction with microtubules of the mitotic kinesin-3 KIF14. KIF14 is involved in cytokinesis. Mutations in the KIF14 gene are associated with developmental diseases, and KIF14 overexpression is associated with cancers. Therefore, the structural study of the mechanism of KIF14 is an important question. This study follows on from a previous study where the structure of the isolated KIF14 motor domain was determined (Arora et al, JMB, 2014). Here, the authors have studied several KIF14 constructs, in different nucleotide states, bound to 15-prot filament microtubules. As much as 18 structures are presented, some of them from dimeric KIF14 constructs. The work focuses mainly on cryo-EM data, with the exception of the oligomeric state of relevant constructs investigated by gel filtration chromatography and of their ATPase activity. This leads to a mechano-chemical model of KIF14 on the microtubule (Fig. 7) which is basically in line with previous models proposed for other kinesins. The main findings of this study are that at least about 15 residues of the neck linker are needed for microtubule-bound ATP-like KIF14 to adopt a closed conformation, and that an ATP nucleotide molecule can bind to each motor domain of a microtubule-bound dimeric construct. Whether these are specific features of KIF14 is not clear.

I acknowledge the amount of cryo-EM data that are presented. However, I have a main difficulty with the data in that there is no estimate of the precision of the coordinates (for instance, as described in Herzik et al, Structure, 2019). The absence of such estimates is problematic. As a first example, Fig. 4c shows differences between the AMP-PNP and AIFx K748 structures. These structures have an RMSD of 0.53 Ang (Table 2). Couldn't these structures be considered very similar (e.g., Hryc et al, PNAS, 2017)? As a second example, the supplementary movie suggests movements within the microtubule part. Are they significant? If so, are they related to the kinesin-induced structural changes of the microtubule (Peet et al, Nat Nanotechnol, 2018)?

Therefore, in the absence of such an estimate, the conclusion of the study should be downplayed and the work seems better suited for a more specialized journal.

Additional comments:

- Fig. 2 is not easy to understand (which is related to my main concern mentioned above). In particular (but not limited to this point), the fact that switch 2 and alpha0 shares the same yellow color does not help.

- Fig. 3 is also difficult to follow. In Fig. 3b, the structures are superposed on the kinesin elements interacting with beta-tubulin. But these elements are not listed. I understand that alpha4 is one of these. It is well established that this helix can move from free kinesin to microtubule-bound kinesin (e.g., see Fig. 4b in Trofimova et al. *Nature commun*, 2018. Also suggested in the case of KIF14 by Fig. 3d). Hence there are probably better choices for superposition than taking alpha4 as a reference. In other words, are we seeing movements between the two sub-domains, or a rearrangement of alpha4 and associated loops upon microtubule binding?

- In addition, Loop 12 (KL12) is mentioned in the text but not labeled. The color code is also not straightforward. For instance, Switch 2 is in yellow in Fig. 2, in magenta in Fig. 2a, and in light pink in Fig. 2d.

- Two sub-domains in KIF14 are identified. They should be better defined, probably with a figure. More importantly, how they compare with the 3 subdomains defined in kinesin-1 (Cao et al, *Nat commun*, 2014; Shang et al, *eLife*, 2014) and further seen in other kinesins (e.g. Atherthon et al, *eLife*, 2017)?

- The KIF14 inter-switch salt bridge is presented as unusual, being also observed in the open conformation (Fig. S6). But from this figure, it is not clear whether the salt bridge is fully formed or only partially formed (with not optimized geometry and distances), hence in this last case not different from the case of kinesin-5 (Parke et al, *JBC*, 2010). An inter-switch interaction (partially formed) was also present in ADP-kinesin-1 not bound to microtubules (Sindelar et al, *NSB*, 2002) and in no-nucleotide kinesin-1 bound to tubulin (Cao et al, *Nat commun*, 2014).

- The difference in the ATPase activity between short and long constructs is likely significant but it is not impressive either (basically a 2-fold difference in the catalytic factor). It would be interesting to record the ATPase activity of a construct with the neck linker fully truncated. In addition, the basal ATPase activity of this kinesin is unusually high (as previously determined by the authors; Arora et al, *JMB*, 2014). In Fig. 5c, is the basal activity subtracted? It would be helpful to give the basal activity for each construct.

- Kinesin constructs are added to microtubules directly on the EM grid. Why is it not done in solution, then the KIF14-decorated microtubules applied on the grid?

- Related to this point, apyrase was previously used to dissociate the nucleotide from the kinesin (Ma and Taylor, *Biochemistry*, 1995). In this case, a 20 min reaction time was needed to complete the reaction. Have the authors checked if the (about) 1 min incubation time leads to nucleotide-free KIF14?

- Comparison between KIF14 and Kinesin-1 (Supplementary Fig. 7). The take-home message from this figure is not straightforward, because of the choice of the authors to take beta-tubulin as a reference and not the kinesins. As far as I can judge from the figure, the differences between the kinesin part are of the same order than those between alpha-tubulin, even when the comparison is with microtubule-bound kinesin-1.

Minor points:

- Benoit et al studied mouse KIF14 whereas disease-associated mutations have been identified in human KIF14. A sequence alignment between mouse and human KIF14 will be welcome.
- The binding affinity of ADP-KIF14 for microtubules is unusually high (Arora et al, JMB, 2014), which allows the authors to determine structures of microtubule-bound ADP-KIF14 constructs. This could be mentioned/reminded.
- Fig. 1C. The MW calibration of the gel filtration column is misleading. Molecules are separated according to their Stokes radius, not to their MW. Actually, the K735 construct (about 38 kDa) migrates as a 50 kDa protein according to this calibration. SEC-MALLS analysis (or the use of another method, e.g. AUC) is required to ascertain the oligomeric state of the constructs in solution.
- Table 3. In the pairs of residues taken to characterize the open/closed state of KIF14, the nature of the residue at position 647 is ambiguous. Is it a Lysine, as stated in Table 3, or a Thr according to the 4ozq structure?
- There are problems with the reference list, where the name of the journal (e.g., Khalil et al; Moawia et al ...) or the volume and page numbers (Reilly et al) are missing. There are also missing references (e.g., Hwang et al., 2017 (page 6) not in the reference list).
- There are many typos and a careful proofreading of the manuscript is needed. As a few examples: swtch-2 (legend to Fig. 2); K35, K43 (Legend to Fig. 5); BL21 pLys: BL21(DE3)pLysS or pLysE, I suppose (Method section); "reactions were assembled in in BRB40-based depolymerization buffer": I do not understand this sentence. In particular, why do the authors use a depolymerization buffer with taxol-stabilized microtubules to record microtubule-enhanced ATP hydrolysis? (Method section, ATPase assay.) "A 3D classification with Frealign in 2 classes to was used" (Method section, 3D reconstruction)...
- The composition of BRB40 is given but not that of BRB80. In particular, is the 1 mM MgCl₂ included in BRB80? It is not clear from the cleavage buffer composition.
- "the microtubule unbound structure": I am not sure we can use such a shortcut.
- In Fig. 4f, the KIF14 motor domain is expected to be in blue. It is not clear what is the helix in magenta. Is that alpha4?
- Fig. 4h-i. Please review the legend. It is not consistent.

We thank the reviewers for their careful consideration of our paper. We have addressed all their major and minor comments and modified the manuscript substantially.

To facilitate revision of the paper, the major changes and additions in the main text are highlighted in blue font. The paper includes a new main figure (new Fig. 7), and five new supplementary figures. Previous Supplementary Fig. 3 was eliminated because of redundancy with other figures. Previous Table 2 was moved to the Supplementary material section (new Supplementary Table 2) to keep the main paper within the 10 maximum number of display items indicated in the Nature Communications author guidelines. Most figures were modified to better document the points made in the text. We also made the abstract more compact (<190 words). Following is our point-by-point responses to each of the reviewer comments (reviewer's comments in bold- italics).

REVIEWER COMMENTS

Reviewer #1 (Remarks to the Author)

This excellent work uses cryo-EM to provide the highest resolution and most comprehensive structural description of a kinesin motor to date. By looking at several different constructs, both monomeric and dimeric, the authors fill in important details of the motility cycle that were previously not well established. An especially noteworthy contribution is the discovery that binding of ATP (modeled by nonhydrolyzable analog AMPPNP) by itself is not sufficient to allosterically transform kinesin to its 'closed', actively hydrolyzing conformation. The experiments here demonstrate this by showing that if the neck linker is truncated, ATP analog binding fails to fully close the nucleotide pocket. Supporting biochemical data are consistent with the structures, showing a somewhat reduced ATPase activity with the truncation.

Overall, the manuscript is clearly written, well organized and the figures are beautifully rendered. I have no major criticisms. I attach a few minor suggestions, but I feel the manuscript could be published essentially as is.

We thank the reviewer for the positive assessment of the work.

P. 5. 'On the other hand, the microtubule bound cryo-EM structures show clear densities along the full length of these regions '-> Ordering of SWI/SWII upon MT or tubulin binding has been observed before in cryo and X-ray, and it would not hurt to cite the previous works that found this (Atherton...Moore, 2013; Gigant...Knossow, 2014; Shang... Sindelar, 2014).

We added the following text to the first paragraph of 'Microtubule binding opens the KIF14 nucleotide binding pocket enabling nucleotide exchange' in results": "This is consistent with previous structural studies that showed ordering of the kinesin loops near the interface with tubulin or the microtubule (Atherton et al., 2014; Shang et al., 2014; Wang et al., 2012)..."

However, note that in most previous cryo-EM structures and in the kinesin-tubulin structures some of the kinesin loops near the interface with tubulin or the microtubule (including the switch loops) are not fully resolved.

P. 7. 'The motor domain in the MT-K735-ANP complex shows a new configuration where the KH0-containing-sub-domain is rotated to an intermediate position between the open and closed configurations'

-> It's not clear to me that this is indeed a new conformation rather than a sum of fractionally occupied open and closed states. The authors could consider referencing Sindelar and Downing, 2010 where evidence was presented that AMPPNP bound kinesin-MT monomers are in equilibrium between open and closed states. The main evidence that MT-K735-ANP is unique conformation seems to come from the density for helix-0 in Supp. Fig3 (green) but I think this kind of density could easily arise from a superposition - resolution is not high enough to say for sure.

Helix-0 is very well resolved in the MT-K735-ANP and other maps (Fig. 2). The pitch of the helix as well as side chains are clearly resolved and we don't see evidence of H0 densities in the 'closed' position in the MT-K735-ANP map. Note that although the resolution achieved in the paper by Sindelar and Downing, 2010 (~9 Å) was state of the art at the time, it was not sufficient to fully resolve the position of H0 or determining conclusively whether there was a mixture of open and closed conformations in the AMP-PNP state. In our case, H0 is fully resolved in the MT-K735-ANP map (overall map resolution 3.1Å) and we don't see evidence for mixed conformations in this particular structure. Note that this is different from other structures where we do see a mixture of conformations in the helically averaged maps (MT-K743-ANP/AAF, MT-K755-ANP/AAF, MT-K772-ANP/AAF). In these cases we see two helix-0 densities corresponding to the open and closed conformations (see new Supplementary Fig. 10). We have now separated the conformations in the helically averaged maps (see response to reviewer 2) and determined that from the monomeric constructs only the partially truncated neck-linker K743 construct show open and closed conformations in equilibrium in the presence of AMP-PNP or ADP-AIF_x. In the case of the K755 and K772 constructs (in the presence of AMP-PNP or ADP-AIF_x) there are also two coexisting conformations corresponding to the leading and trailing heads that we have now separated. With the exception of the MT-K743-ANP/AAF datasets all the monomeric constructs including the MT-K735-ANP produces a single conformation.

P. 10. 'Consistent with the structural results, we found that the K735 and K743 constructs have reduced microtubule stimulated ATPase activity relative to the longer KIF14 constructs'. -> I encourage the authors to reference and cite Gigant...Knossow, 2014, here, as they found similar (actually, much greater) effects in a truncated neck linker mutant using ATPase assays.

We now cite in this paragraph Cao/Wang/Jiang/Wang/Knossow/Gigant 2014 and also Nitta et al., 2008.

MINOR EDITS:

*P. 5: observed with the *construct* that include at least 15 neck linker residues
-> constructs
Fixed.*

*P. 12; even though only the K772 *constructs*
-> construct
Fixed.*

*p. 15: The fact neck-linker docking -> "fact that
Fixed.*

Reviewer #2 (Remarks to the Author):

This paper by Benoit et al describes the cryo-EM structures of mitotic kinesin, KIF14. They used five constructs with different length (they all have full-length motor domain and have different length of neck linker or dimerization coiled-coil region). Authors took EM data sets in four different nucleotide state (ADP, apo, AMPPNP, and ADPAIFx) for each construct. In total, they solved 18 unique EM maps at 2.9 to 4 Å resolution and made 22 unique atomic models. The cryo-EM analysis is technically sound and quite extensive.

Using these structures, the authors discuss about the ATP-dependent structural changes in KIF 14 i.e. (1) Structural changes of KIF14 induced by microtubule binding. (2) Structural changes after ATP binding and followed by its hydrolysis, (3) Allosteric regulatory mechanisms for the structural changes involved in hydrolysis by neck linker docking. (4) The mechanochemical cycle of dimer KIF14 based on their dimer atomic models. The most important point in this paper is that they have determined the structure of the motor domain-microtubule complex with various lengths of constructs in various nucleotide states, and they classified them into 4 group (semi-closed, open, open and closed), allowing them to*

discuss in detail about the nucleotide-dependent conformational changes in KIF14. It is particularly valuable that their new ATP or ADP Pi analogue structure from truncated neck linker constructs allow them to discuss about the allosteric regulatory mechanisms for the structural changes involved in hydrolysis by neck7/22/2020 Mail - Hernando J Sosa - Outlook <https://outlook.office365.com/mail/deeplink?version=2020071201.02&popoutv2=1> 5/8 linker docking. The reviewer think that their experimental results and the discussion about the monomeric KIF14 motor domain are of general interests to biophysicists studying biological motors.

We thank the reviewer for the positive assessment of the work.

On the other hand, their structural model of the dimer are not consistent with the current understanding of conventional kinesin movements. Therefore, it require additional experimental data. Therefore, the reviewer suggests revising the interpretations and models of dimer KIF14.

Major points:

The authors created dimer models and used it as the basis of Movie 2. However, it require more experimental data to support the model shown in the movie.

(1) If the authors want to support the model based on structural studies, they have to distinguish the leading and trailing head of KIF14 as Liu et al (2017) did for conventional kinesin. In the current manuscript, the 3D reconstruction was done assuming that there are no differences in the leading and trailing heads. Therefore, the current 3D reconstructions of K772 are the average of the leading and trailing heads. The authors derived two possible models of the dimer structure from a single averaged structure map.

At the resolution of the helically averaged maps it was clear that there were two motor domain averaged structures in the MT-K755-ANP/AAF and MT-K772-ANP/AAF maps and that they matched the open and closed conformations obtained from the monomeric constructs (e.g. MT-K748-ADP and MT-K748-ANP see new Supplementary Fig. 10). This and the results with the constructs with truncated neck linker led to the proposed dimeric structures in the two-heads-bound state with leading and trailing motor domains in distinct configurations. Note also that we did not propose 'two possible' models of the dimer structure but rather a dimer in which the two motor domains are in distinct conformations, one in the open conformation (leading head) and the other in the closed conformation (trailing head). However, we agree with the reviewer in that that distinguish the leading and trailing head would be ideal to support the proposed model.

Obtaining a 3D reconstruction of a microtubule bound kinesins dimer where the densities corresponding to the leading and trailing heads are separated and not averaged is a difficult problem first addressed by Liu et al (2017). However, without taking merit of the excellent and important paper by Liu et al it is important to note that the resolution achieved by Liu et al is much less than our helically averaged maps and insufficient to distinguish between the open and closed conformations. We have been working on the problem of separating coexisting conformations in helical assemblies for a while and we are now including the results in this revised version (new Fig. 7 and Supplementary Fig 2). To separate the two heads of microtubule bound dimers we implemented an image analysis procedure that takes advantage of recent computational capabilities present in the single particle analysis software package Relion (partial signal subtraction, symmetry expansion and 3D classification). We called this procedure Helical Assembly Subunit Refinement and Classification (HASRC). Important differences with the procedure used by Liu et. al, is that our procedure can be applied to fully decorated microtubules and uses unbiased classification. In addition, the resolution achieved with the HASRC method is even higher than the corresponding helically averaged maps. Using this method, we have now produced the first near atomic resolution structures of microtubule bound kinesin dimers in the two-heads bound state. The structures confirm what we previously concluded from the helically averaged maps; that the two motor domains were in distinct conformations with leading and trailing heads in the open and closed conformation respectively. The new maps also reveal important differences from the open and closed structures of the monomeric constructs that were previously obscured in the average.

(2) The authors do not consider the possibility of artifacts due to the excess amount of nucleotides to the solution.

In fact, the authors present a structure in which ADP-AlFx (ADP Pi analog) is attached to both heads and both heads take open conformations in movie2. Such structures are achieved only in mutants with extended neck linker, but not in the wild-type for conventional kinesin, as demonstrated by single-molecule measurements (Andreasson et al 2015) from Block lab. It should be shown whether the dimer models they present in this paper is possible in the case of KIF14.

References

Structural basis of cooperativity in kinesin revealed by 3D reconstruction of a two-head-bound state on microtubules.

Liu, D., Liu, X., Shang, Z., Sindelar, C. V.

Elife 2017 6:

Examining kinesin processivity within a general gating framework.

Andreasson, J. O., Milic, B., Chen, G. Y., Guydosh, N. R., Hancock, W. O., Block, S. M.

Elife 2015 4:

In the case of the ATP analogue AMP-PNP the concentration used (4 mM) is within the physiological ATP concentrations (1-10 mM) and therefore the results (nucleotide site occupancy) are also likely to reflect what happens *in-vivo*. In the case of ADP-AIF_x the concentrations are relatively high and the structures we solved with these nucleotides are interpreted as transient structures produced when the active state contains ADP-Pi. We do not propose an intermediate in the KIF14 dimer mechanochemical cycle were ADP-Pi (as mimicked by ADP-AIF_x) is present in both the leading and trailing heads. Note also that if the hydrolysis step is slowed down in the leading head (open conformation) as we propose, the likelihood of finding this head in the same post-hydrolysis state of the trailing head is low. There was an intermediate in movie 2 (but not in the cartoon show in Fig. 7, now Fig. 8) where the leading head is AMP-PNP bound and open and the trailing head is ADP-AIF_x bound and open (i.e., an open-open structure as the reviewer indicates). We do not have strong evidence for this particular intermediate and therefore we have now eliminated it from Supplementary Movie 2.

Reviewer #3 (Remarks to the Author):

Benoit et al present a cryo-EM study of the interaction with microtubules of the mitotic kinesin-3 KIF14. KIF14 is involved in cytokinesis. Mutations in the KIF14 gene are associated with developmental diseases, and KIF14 overexpression is associated with cancers. Therefore, the structural study of the mechanism of KIF14 is an important question. This study follows on from a previous study where the structure of the isolated KIF14 motor domain was determined (Arora et al, JMB, 2014). Here, the authors have studied several KIF14 constructs, in different nucleotide states, bound to 15-protofilament microtubules. As much as 18 structures are presented, some of them from dimeric KIF14 constructs. The work focuses mainly on cryo-EM data, with the exception of the oligomeric state of relevant constructs investigated by gel filtration chromatography and of their ATPase activity. This leads to a mechano-chemical model of KIF14 on the microtubule (Fig. 7) which is basically in line with previous models proposed for other kinesins. The main findings of this study are that at least about 15 residues of the neck linker are needed for microtubule-bound ATP-like KIF14 to adopt a closed conformation, and that an ATP nucleotide molecule can bind to each motor domain of a microtubule bound dimeric construct. Whether these are specific features of KIF14 is not clear.

I acknowledge the amount of cryo-EM data that are presented. However, I have a main difficulty with the data in that there is no estimate of the precision of the coordinates (for instance, as described in Herzik et al, Structure, 2019). The absence of such estimates is problematic.

The structures we present in this paper are at the higher resolution of any kinesin-microtubule complex to date (now even higher in this new version). We provided all the parameters derived from the cryo-EM structural analysis and modeling according to current standards established in the cryo-EM field (Table 1, Supplementary Table 1 and Supplementary Figure 2) but we thank the reviewer for suggesting the addition of the novel precision metric as described by Herzik et al, (Structure, 2019). The novel metric and related figures are now included in the new Supplementary Figure 5. The analysis shows that the coordinates precision according to the Herzik et. al. estimate (per residue C α RMSD between fitted models) in almost the entirety of all the structures we provide is ≤ 1 angstrom which is more than sufficient to support the significance of the structural changes we describe in the paper.

As a first example, Fig. 4c shows differences between the AMP-PNP and AIFx K748 structures. These structures have an RMSD of 0.53 Ang (Table 2). Couldn't these structures be considered very similar (e.g., Hryc et al, PNAS, 2017)?

The structures can be considered very similar and this is indeed what we wanted to convey in the paper, both of these structures are in the 'closed' conformation. However, although the structures are very similar they are not identical. Most of the difference is due to the fact that there is a slight rotation of part of the motor domain between the two structures. This relatively small and localized difference is minimally reflected in the overall RMSDs calculated in Table 2 (now Supplementary Table 2). The structural difference, although small, is reproducible between four independent pairs of ANP/AAF structures. It is also higher than the deviation between the models used to calculate the coordinates precision (new Supplementary Fig. 9).

As a second example, the supplementary movie suggests movements within the microtubule part. Are they significant? If so, are they related to the kinesin-induced structural changes of the microtubule (Peet et al, Nat Nanotechnol, 2018)?

The focus of this paper is in the structural changes induced by nucleotides and microtubule binding on the structure of KIF14 and how they relate to the mechanochemical cycle of motile kinesins. We highlight some observed differences in tubulin structure (e.g. the different rotameric position of the side of alpha-tubulin-Y108, Fig. 4) but we considered out of the scope of the present paper an in-depth-analysis of the effect of KIF14 and other kinesins, such as kinesin-1 (as used in Peet et al. 2018) on MT structure.

Therefore, in the absence of such an estimate, the conclusion of the study should be downplayed and the work seems better suited for a more specialized journal.

The required estimate is now provided (new Supplementary Fig. 9) and it further supports the conclusions of the paper (see first response to the reviewer comment above).

Regarding the significance of this paper to the wider Nature Communications audience this paper addresses long-standing questions in biological molecular motors of interest across multiple fields, such as cell biology, biochemistry, biophysics and nano-technology. Our paper for the first time provides a near atomic resolution view of all the key conformational changes of a molecular motor bound to its filament track. Our study separates the effects of microtubule binding, the nucleotide species in the active site, the neck-linker configuration and dimerization on the structure of the kinesin motor domain(s) at near atomic resolution.

Additional comments:

- Fig. 2 is not easy to understand (which is related to my main concern mentioned above). In particular (but not limited to this point), the fact that switch 2 and alpha0 shares the same yellow color does not help.

We apologize for the color selection. Figure 2 is now re-colored to make it consistent with Figure 3. We added labels to highlight the different conformations of the KIF14 motor domain (closed or open) and the position of the neck-linker domain. We also now display all the views at the same scale.

-Fig. 3 is also difficult to follow. In Fig. 3b, the structures are superposed on the kinesin elements interacting with beta-tubulin. But these elements are not listed. I understand that alpha4 is one of these. It is well established that this helix can move from free kinesin to microtubule-bound kinesin (e.g., see Fig. 4b in Trofimova et al. Nature commun, 2018. Also suggested in the case of KIF14 by Fig. 3d). Hence there are probably better choices for superposition than taking alpha4 as a reference. In other words, are we seeing movements between the two sub-domains, or a rearrangement of alpha4 and associated loops upon microtubule binding?

We apologize for the omissions related to Fig. 3b. The regions (KL8 and KL12) are now listed in the legend. Figure 3b has also been modified adding insets for all the KIF14 microtubule interacting regions.

We show two possible alignments or superpositions to highlight different aspects of the conformational changes induced by microtubule binding. We align to the β -tubulin interacting

regions to highlight the fact that microtubule binding induces a displacement between the KIF14 microtubule interacting regions (Fig.3b & c). We align to the P-loop (Fig. 3d) to highlight the conformational change (opening) of the nucleotide binding site. Replacing the alignment in Figures 3b and 3c for H4 would result in very similar figures, as the orientation of H4 changes little *relative to the β -microtubule binding site*. We do see a small change in the orientation of H4 *relative to the P-loop and other parts of the kinesins motor domain* as shown in Fig. 3d.

Note that when the reviewer refers to Fig. 4b of Tromifova et. al. the structures being compared there are of a microtubule unbound kinesin in the ADP state with the structure of a similar kinesin bound to curved tubulin (not a microtubule) in the presence of AMP-PNP. Therefore, the structural differences observed between these structures are related to the binding/unbinding to curved tubulin, and/or whether ADP or AMP-PNP is in the active site. This is very different to the case shown in Figure 3 of our paper where two structures, in the same nucleotide state (ADP) but differing in whether they are microtubule bound or unbound, are compared. This in general is a strong point of our paper that perhaps was not made sufficiently clear. We completely separate the effects of microtubule binding, nucleotide species in the active site and neck-linker configuration on the structure of a kinesin motor domain.

We should also point out that an earlier study by our group (Benoit et al., 2018) showed that a similar kinesin (kinesin-13) as used by Tromifova et al. only undergoes a conformational change as depicted in Tromifova et al. Fig. 4b when bound to curved tubulin but not when bound to the microtubule. Thus, it is not always the case as the reviewer indicates that "***It is well established that this helix can move from free kinesin to microtubule-bound kinesin (e.g., see Fig. 4b in Trofimova et al. Nature commun, 2018)***"

In relation to "***are we seeing movements between the two sub-domains, or a rearrangement of alpha4 and associated loops upon microtubule binding?***"

We did describe the conformational change 'to a first approximation' as a movement between sub-domains (Fig. 4a-c. Supplementary Fig. 7, Supplementary Movie 1). The movement cannot be simply described as a rearrangement of alpha-4.

- In addition, Loop 12 (KL12) is mentioned in the text but not labeled. The color code is also not straightforward. For instance, Switch 2 is in yellow in Fig. 2, in magenta in Fig. 2a, and in light pink in Fig. 2d.

KL12 it is not labeled in Fig. 2 but it is labeled in Fig. 3 (first KL12 reference in text refers to Fig. 3). Switch-2 and KH4 are now colored in magenta in Figs. 2, 3a and 3d.

- Two sub-domains in KIF14 are identified. They should be better defined, probably with a figure. More importantly, how they compare with the 3 subdomains defined in kinesin-1 (Cao et al, Nat commun, 2014; Shang et al, eLife, 2014) and further seen in other kinesins (e.g. Atherthon et al, eLife, 2017)?

A new figure (Supplementary Fig. 7) has been added where the subdomains are outlined and compared with the subdomains defined by Cao et. al., 2014. Note that any subdomain-definition would depend on the arbitrary selection of the tolerance for displacement allowed within any part of the protein to be considered a 'rigid' sub-domain. This is the reason why we used the qualifier 'to a first approximation' when describing the subdomains definitions. We color coded and used vectors to highlight the magnitude and direction of the displacements between equivalent residues in the two conformations compared. We believe this provide a more complete description of the structural differences in different regions of the motor domain than a pseudo-arbitrary definition of sub-domain boundaries.

There is some overlap between the KIF14 subdomains we describe and the domains defined by Cao et al., but they are not identical (See new Supplementary Fig. 7). Regarding whether the same subdomains as described by Cao. et al. have been "*further seen in other kinesins (e.g. Atherton et al, eLife, 2017)*", this is not the case. Almost all previous structural work on kinesin-microtubule complexes are at $>5 \text{ \AA}$ overall resolution (and lower resolution on the kinesin part), including Atherton et al, eLife, 2017 (5.5-8 \AA). At these resolutions it is difficult to assess whether parts of the motor domain move together as rigid bodies or not. Also, the final 'atomic' models are more biased towards the initial models used when fitting the structures to the experimentally determined densities. Atherton et al., used the coordinates of Cao e et al. (4LNU & 4HNA) as initial templates during fitting and they assigned the same subdomains as Cao e al., adding an MKLP specific subdomain (From Atherton et al., Fig 2: "Subdomains for kinesin-1 and MKLP2-MD are assigned and colored based on Cao et al"). Thus, the subdomains described by Cao et al were not **independently** found by Atherton et al. This discussion highlights one aspect of why determining the structure of kinesin-microtubule complexes at better resolution than previously available, as done in the present paper (< 4 angstroms), is of significance.

One important insight we derive from our analysis is that we recognize that there is no simply one tubulin binding subdomain. Instead, the microtubule binding areas are split into the three subdomains that we now call the plus, minus and switch subdomains. At the interface between these domains locate the nucleotide and neck-linker binding pockets. This, we propose is a key factor coupling microtubule binding with the kinesin mechano-chemical cycle. A new section was added in the discussion to cover these points.

- The KIF14 inter-switch salt bridge is presented as unusual, being also observed in the open conformation (Fig. S6). But from this figure, it is not clear whether the salt bridge is fully

formed or only partially formed (with not optimized geometry and distances), hence in this last case not different from the case of kinesin-5 (Parke et al, JBC, 2010). An inter-switch interaction (partially formed) was also present in ADP-kinesin-1 not bound to microtubules (Sindelar et al, NSB, 2002) and in nonucleotide kinesin-1 bound to tubulin (Cao et al, Nat commun, 2014).

We have now eliminated from the text the general claim that we observed this salt-bridge in an 'open-conformation' for the first time. We modified the text to better describe the novelty of our findings. We cannot be certain of the exact orientation of the side chains of the residues forming the salt bridge, and indeed the situation may be similar to Parke et al., JBC 2010. What we show for the first time in a *microtubule-bound-kinesin* is the full path of the polypeptide chain forming the two switch loops including the densities associated with the side chains of the two residues implicated in forming the salt bridge between them (Supplementary Fig. 6). The cryo-EM densities and fitted models show that the side chains of these two residues point to each other and are within the distance required to form a salt bridge. This we believe is the best evidence so far indicating that this salt bridge, which has been argued to be important for catalysis, forms when the motor domain is bound to the microtubule (in the open or closed conformation).

- The difference in the ATPase activity between short and long constructs is likely significant but it is not impressive either (basically a 2-fold difference in the catalytic factor). It would be interesting to record the ATPase activity of a construct with the neck linker fully truncated. In addition, the basal ATPase activity of this kinesin is unusually high (as previously determined by the authors; Arora et al, JMB, 2014). In Fig. 5c, is the basal activity subtracted? It would be helpful to give the basal activity for each construct.

Our K735 construct is equivalent to the 'fully' truncated neck-linker kinesin-1 construct used by Cao et al., 2014 who reported a larger reduction in ATPase activity due to the neck-linker truncation. Therefore, differences in ATPase activities between the truncated constructs in Cao et al., 2014 and our paper are most likely related to the different kinesin species used (KIF14 vs. KIF5b). Note that whether the neck linker is fully truncated or almost fully truncated depends on what is taken as the last residue of KH6 and the first residue of the neck linker. We originally took the first neck-linker as Arg-734 for consistency with some papers in the literature. KIF14 Arg-734 aligns with a lysine in other kinesins than in previous papers has been labeled as the first residue of the neck linker (e.g. Shastry & Hancock, Current Biology 2010). However, in other papers (Gigant et al Nat. Struct & Mol Biol 2013, Cao et al Nat. Comm 2014) the first residue of the neck linker of the kinesin-1 KI5b is considered to be Ile-325 which aligns with KIF14 Ile-736. We have now changed the numbers through the text according to this definition (i.e. KIF14 Ile-736 as the first neck-linker residue).

The basal activity has been subtracted from the data plotted. This is now indicated in the Fig. 5c legend together with the basal rate values.

- *Kinesin constructs are added to microtubules directly on the EM grid. Why is it not done in solution, then the KIF14-decorated microtubules applied on the grid?*

We have done it both ways but found improved decoration of microtubules when kinesins is added to the grid.

- *Related to this point, apyrase was previously used to dissociate the nucleotide from the kinesin (Ma and Taylor, Biochemistry, 1995). In this case, a 20 min reaction time was needed to complete the reaction. Have the authors checked if the (about) 1 min incubation time leads to nucleotide-free KIF14?*

The best evidence that the concentration of apyrase and the incubation time resulted in nucleotide-free KIF14 is that the structures in the Apo conditions are nucleotide free. There is no nucleotide associated densities in the active site of the structures obtained in the 'Apo' conditions. (Fig. 5d and Supplementary Fig. 11). We used an excess of apyrase that according to the manufacturer's specifications should be sufficient to remove all traces of ADP or ATP during the incubation time.

Note that this is another aspect where the higher resolution of the kinesin-microtubule structures obtained in the present study is important: the nucleotide species present (or missing) in the kinesin active site can be determined directly.

- *Comparison between KIF14 and Kinesin-1 (Supplementary Fig. 7). The take-home message from this figure is not straightforward, because of the choice of the authors to take beta-tubulin as a reference and not the kinesins. As far as I can judge from the figure, the differences between the kinesin part are of the same order than those between alpha-tubulin, even when the comparison is with microtubule bound kinesin-1.*

We apologize for the lack of clarity here. The take home message of this figure is that the structures of the *complexes*, not the individual chains, are different. We have modified the Figure (now Supplementary Fig. 12) and legend to better explain this.

All the kinesins motor domain structures compared in that figure are relatively similar (all in an open conformation) except for the regions that are not resolved (and presumed disordered) in the kinesin-1 structures but that are resolved in the KIF14 structures (now highlighted in a different color). However, the orientation/position of the motor domain relative to the tubulin chains to which they are bound are different between the kinesin-1 and the KIF14 complexes. We aligned

all the complexes to the beta-tubulin subunit to highlight this. Note that with this alignment relatively large differences are also expected to be seen in the alpha-tubulin part of the complex when comparing the kinesin-1-curved-tubulin complex (4LNU) with the KIF14-microtubule structures due to the different tubulin conformations in these complexes (curved tubulin vs. straight tubulin in the microtubule). Note also that the difference between kinesin-1 and the KIF14 complexes is larger than the differences between the four independent MT-KIF14-Apo structures presented.

Minor points:

- Benoit et al studied mouse KIF14 whereas disease-associated mutations have been identified in human KIF14. A sequence alignment between mouse and human KIF14 will be welcome.

A new Supplementary Fig. 1 was added with the sequence alignment.

- The binding affinity of ADP-KIF14 for microtubules is unusually high (Arora et al, JMB, 2014), which allows the authors to determine structures of microtubule-bound ADP-KIF14 constructs. This could be mentioned/reminded.

A sentence was added at the end of the second paragraph of section 'Cryo-electron microscopy structures of KIF14 motor domain complexes' in Results mentioning this fact.

- Fig. 1C. The MW calibration of the gel filtration column is misleading. Molecules are separated according to their Stokes radius, not to their MW. Actually, the K735 construct (about 38 kDa) migrates as a 50 kDa protein according to this calibration. SEC-MALLS analysis (or the use of another method, e.g. AUC) is required to ascertain the oligomeric state of the constructs in solution.

We added a clarification to the legend of Fig. 1C and modified the related text in the first paragraph of results accordingly. Note that we are not claiming that we know the oligomeric state of the constructs just with this experiment. The fact that the K772 construct migrates significantly faster than the shorter constructs suggests that it forms a higher oligomer in solution. Given that the difference between constructs is the presence of a coiled-coil domain which in most kinesin lead to the formation of a dimer, the most likely conclusion is that the K772 construct dimerizes in solution. The EM data shows directly that all the construct without coiled-coil CC1 or part of it (K748 and shorter) bind to the microtubule as monomers while the constructs that include part of CC1 form dimers when microtubule bound (visualized in the AMP-PNP and ADP-AIF_x two-heads-bound states).

- Table 3. In the pairs of residues taken to characterize the open/closed state of KIF14, the nature of the residue at position 647 is ambiguous. Is it a Lysine, as stated in Table 3, or a Thr according to the 4ozq structure?

K647 in Table 3 is a typo. It should be T647 as pointed out by the reviewer, it is now corrected. We thank the reviewer for pointing this out.

- There are problems with the reference list, where the name of the journal (e.g., Khalil et al; Moawia et al ...) or the volume and page numbers (Reilly et al) are missing. There are also missing references (e.g., Hwang et al., 2017 (page 6) not in the reference list).

Fixed. We thank the reviewer for pointing out these errors.

- There are many typos and a careful proofreading of the manuscript is needed. As a few examples: swtch-2 (legend to Fig. 2); K35, K43 (Legend to Fig. 5); BL21 pLys: BL21 (DE3) pLysS or pLysE, I suppose (Method section): “reactions were assembled in BRB40-based depolymerization buffer”: I do not understand this sentence. In particular, why do the authors use a depolymerization buffer with taxol stabilized microtubules to record microtubule-enhanced ATP hydrolysis? (Method section, ATPase assay.) “3.13b) A 3D classification with Frealign in 2 classes to was used” (Method section, 3D reconstruction).

Typos and erroneous buffer names ('depolymerization') have been fixed (pLysS is correct). We thank the reviewer for the careful proofreading.

- The composition of BRB40 is given but not that of BRB80. In particular, is the 1 mM MgCl2 included in BRB80? It is not clear from the cleavage buffer composition

Both contain 1mM MgCl₂. The full composition of the BRB80 buffer is now stated.

- “the microtubule unbound structure”: I am not sure we can use such a shortcut.

We used the above shortcut to stress the fact that we are comparing the structures of kinesin motor domains when bound or not-bound to the microtubule.

- In Fig. 4f, the KIF14 motor domain is expected to be in blue. It is not clear what is the helix in magenta. Is that alpha4?

KH4 and SW2 are in magenta in Fig. 4c-j, this is now stated in the legend.

- Fig. 4h-i. Please review the legend. It is not consistent.

Inconsistencies in the Fig.4 legend have been fixed.

Reviewer #1 (Remarks to the Author):

Although I have one reservation (see below), this revised version of the manuscript is significantly improved from the previous version, especially due to the successful structure determination of a two-heads bound kinesin motor state at near-atomic resolution. The new results did not greatly affect the overall message of the paper, although it is noted that one of the former claims- that aluminum fluoride could be visualized in the lower half of the open nucleotide pocket of the leading head- seems to have been revised.

Overall, the new results, discussions and supplementary figures all add to the value of the paper. My only remaining concern is that the novelty seems to be overstated, and references missing, at two points in the paper.

First, in the new discussion section 'Microtubule and nucleotide induced subdomain movements in the kinesin motor domain' there is a curious omission of the model from Shang et al. (Ref. 22). While the text here gives the impression that the 'two-domain' model illustrated in Supplementary Fig. 7 is new, it can be compared directly to the 'clamshell' model presented in Shang et al. (Fig. 8) which presents a detailed structural argument not only for why kinesin's microtubule-bound behavior approximates a two-domain movement, but also for why the third subdomain (there called the 'lower' subdomain, equivalent to the 'MT-binding subdomain' of Ref. 18) becomes articulated in kinesin's detached states. This reference should be cited accordingly and that model contrasted with observations in the current paper- the main difference that I can see being that Shang et al. did not anticipate nucleotide binding to the leading head (although, as noted in this manuscript, the same group later observed evidence for this in their 2017 kinesin dimer structure paper, Ref. 48).

Second, the 'HSARC' method is presented without any reference to Debs et al., 117:16976–16984 PNAS 2020, which presented essentially the identical method with similar results- improving microtubule resolution and classifying different kinesin binding states. Again, this reference should be included and any differences/improvements of the HSARC method noted.

Otherwise, I think this paper would now be suitable for publication; my congratulations to the authors for a nice piece of research.

Reviewer #2 (Remarks to the Author):

They made the following improvements to the two major issues we presented.

- Overcoming problems with the reconstruction of dimer constructs

We pointed out that the structure produced by helical reconstruction from dimer constructs is a mixture of leading and trailing head. Then we recommended the method of Liu et al. for reconstructing dimers, in which the front and back head can be distinguished within a reconstructed structure. Beyond our expectations, the authors devised a new method (HASRC) in this revision that is superior to the method of Liu et al. in terms of resolution, and presented several dimer structure (supplemental Fig. 2.3). This is an adaptation of the classification method implemented in Relion 3.1. By applying this new method, they were able to improve the dimer structure (resolution) up to 3.1

Å. In addition, the resolutions of the monomers KIF14 were also improved by up to 2.7 Å. Because of this improvement in resolution, the authors show the difference between open and closed conformation more clearly.

- Correction of problems with the dimer kinetic model

In the previous version, the authors proposed a dimer kinetic model in which an open-open intermediate appears. Since such an intermediate is rejected in conventional kinesin, the reviewer commented that supporting data should be presented if it is to be proposed as an intermediate state unique to KIF14. Since there is no strong evidence to suggest the existence of such intermediates, in this revision, the authors modified a part of their proposed model and excluded the part where this particular model appears from Movie2. This modified model seems to work well in the current situation for KIF14.

The author has responded well to other reviewer's points and requests, and has made it more understandable for general reader. I think the revised manuscript is suitable for publication in Nature communication.

Reviewer #3 (Remarks to the Author):

The authors have responded reasonably to comments on the previous version of this manuscript. I support publication without further change.

We thank again the reviewers for their carefully reading of our manuscript and we are happy to know that they all now consider the paper acceptable for publication in Nature Communications. We also wish to thank the reviewers for their constructive input. We have modified the document to address the editorial requests and the reservations indicated by reviewer-1 (changes in main text highlighted in blue fonts). Following is the point-by-point responses to the reviewers.

Reviewer #1 (Remarks to the Author):

Although I have one reservation (see below), this revised version of the manuscript is significantly improved from the previous version, especially due to the successful structure determination of a two-heads bound kinesin motor state at near-atomic resolution. The new results did not greatly affect the overall message of the paper, although it is noted that one of the former claims- that aluminum fluoride could be visualized in the lower half of the open nucleotide pocket of the leading head- seems to have been revised.

Overall, the new results, discussions and supplementary figures all add to the value of the paper. My only remaining concern is that the novelty seems to be overstated, and references missing, at two points in the paper.

First, in the new discussion section 'Microtubule and nucleotide induced subdomain movements in the kinesin motor domain' there is a curious omission of the model from Shang et al. (Ref. 22). While the text here gives the impression that the 'two-domain' model illustrated in Supplementary Fig. 7 is new, it can be compared directly to the 'clamshell' model presented in Shang et al. (Fig. 8) which presents a detailed structural argument not only for why kinesin's microtubule-bound behavior approximates a two-domain movement, but also for why the third subdomain (there called the 'lower' subdomain, equivalent to the 'MT-binding subdomain' of Ref. 18) becomes articulated in kinesin's detached states. This reference should be cited accordingly and that model contrasted with observations in the current paper- the main difference that I can see being that Shang et al. did not anticipate nucleotide binding to the leading head (although, as noted in this manuscript, the same group later observed evidence for this in their 2017 kinesin dimer structure paper, Ref. 48).

It can be said that closing of the nucleotide binding pocket in microtubule bound KIF14 can be described at low resolution by a clamshell-seesaw type movement as described by Shang et al. 2014 (ref 22) and Sindelar et al, 2011. We have now added a sentence in the discussion citing these two papers with the second reference added to the reference list. Besides this similarity the model differs in several significant points from ours. Enumerating the differences of this particular model and ours in the manuscript would have required adding a special lengthy discussion that we felt was not justified. However, we have expanded on the most significant differences here:

1) In Shang et al the mechanism of coupling between microtubule binding and the nucleotide-pocket structure involves a *linchpin* residue, N255 (equivalent to N662 in KIF14) and an opening/closing of a *polymer-cleft* between helix-4 and SW-II as shown in Fig. 3 of Shang et al. However, we don't single out

a single residue (N662) as the key to explain how microtubule-binding is coupled to the nucleotide binding pocket. We don't see an opening or closing of a polymer cleft in this area as described by Shang et al when comparing any of the KIF14 structures in any of the three conformational states identified (semi-closed, open or closed). Also, the position of the switch-II loop relative to N662 changes little in comparison to other conformational changes between KIF14 microtubule bound and unbound structures. This makes N662 unlikely to be the main driver or the main trigger of conformational changes in the nucleotide binding pocket linked to microtubule binding. On the other hand, a major structural change that we observe between the KIF14 microtubule bound and unbound structures or open vs closed structures is the rotation of a region or subdomain relative to other subdomains. Because the nucleotide binding pocket is at the interface between these subdomains, their relative rotation alters its structure (open->closed->semi-closed). Our model highlights not one but many residues involved in defining distinct microtubule interfaces. In particular the change in relative position between residues in L8, L12, H4 located in what we call the plus subdomain and residues in H6 and L2 in what we call the minus subdomain. In Shang et al. H6 is located in a subdomain that does not interact with the microtubule (called the N-terminal domain) or their potential interactions with the microtubule are not deemed as relevant to the coupling mechanism.

2) We based our model in the comparison of kinesins structures in several states including the comparison between kinesin-ADP-microtubule-unbound vs. kinesins-ADP-microtubule-bound. This comparison is key to understand the effect of microtubule binding on kinesin structure. The model of Shang et al. on the other hand, is based in the comparison of kinesin ADP or AMP-PNP microtubule unbound structures with Apo and ADP-AIFx microtubule bound structures.

3) The structures solved in Shang et al are at $\sim 6\text{\AA}$ resolution. At his resolution side chains, individual beta strands, parts of the polypeptide chain and specific contacts between residues are not resolved. As a consequence, many of the details of their proposed model are necessarily speculative. This also applies to other models in the literature derived from cryo-EM data at similar or less resolution. On the other hand, the model we propose is based on the comparison of structures where these parts are resolved.

4) Our subdomain division, although recognized as an approximation, is based on the measured displacements between equivalent alpha-carbons in pairs of structures or using a subdomain finding algorithm, as explained in Supplementary Figure 7. The subdomains in Shang et al. appear assigned ad-hoc or based in the ones proposed by Cao et al, which we discuss in Supplementary Fig. 7. In any case the three subdomains defined by Shang et al. are not the same subdomains defined in our paper. Among the differences we included the switch loops and only the switch loops in a separate subdomain (the SW subdomain); also in Shang et al., two of the sub-domains interact with the microtubule while in our proposed model the three subdomains form part of the kinesin-microtubule interface. This we argued is a key aspect of the coupling mechanism between the kinesin mechanochemical cycle and microtubule binding.

Second, the 'HSARC' method is presented without any reference to Debs et al., 117:16976–16984 PNAS 2020, which presented essentially the identical method with similar results- improving microtubule resolution and classifying different kinesin binding states. Again, this reference should

be included and any differences/improvements of the HSARC method noted.

We now cite Debs et al. PNAS 2020 in the methods. The method in this paper and ours use Relion symmetry expansion and signal subtraction, procedures that were used or introduced earlier in the two cited papers (Bai et al., 2015 and Ilca et al. 2015). However, the method of Debs et al., and ours were developed independently and address different problems. The method in Debs et al., finds and correct variability in the relative position between protofilaments in pseudo-helical microtubules (microtubules with seams). Our method was designed to refine and classify the subunits (where a subunit is defined as one or two contiguous tubulin heterodimers with associated kinesin motor domains) in helical microtubules. Although the method of Debs et al. is able to separate kinesin bound/unbound states, this is different than classifying coexisting kinesin motor domain conformations and refine them to high resolution. Our method implements specific subunit location refinement and classification steps that were essential to separate coexisting kinesin motor domain conformations (not only kinesin bound/unbound states) at high-resolution.

Otherwise, I think this paper would now be suitable for publication; my congratulations to the authors for a nice piece of research.

We thank the reviewer his positive evaluation of the work and for his input in this and the previous version of the paper.

Reviewer #2 (Remarks to the Author):

They made the following improvements to the two major issues we presented.

- *Overcoming problems with the reconstruction of dimer constructs*

We pointed out that the structure produced by helical reconstruction from dimer constructs is a mixture of leading and trailing head. Then we recommended the method of Liu et al. for reconstructing dimers, in which the front and back head can be distinguished within a reconstructed structure. Beyond our expectations, the authors devised a new method (HSARC) in this revision that is superior to the method of Liu et al. in terms of resolution, and presented several dimer structure (supplemental Fig. 2.3). This is an adaptation of the classification method implemented in Relion 3.1. By applying this new method, they were able to improve the dimer structure (resolution) up to 3.1 Å. In addition, the resolutions of the monomers KIF14 were also improved by up to 2.7 Å. Because of this improvement in resolution, the authors show the difference between open and closed conformation more clearly.

We appreciate the reviewer comment here. Solving the problem of separating mixed conformations was indeed a lot work, but thanks to the reviewer request the paper story has now been considerably strengthened.

- *Correction of problems with the dimer kinetic model*

In the previous version, the authors proposed a dimer kinetic model in which an open-open intermediate appears. Since such an intermediate is rejected in conventional kinesin, the reviewer commented that supporting data should be presented if it is to be proposed as an intermediate state unique to KIF14. Since there is no strong evidence to suggest the existence of such intermediates, in this revision, the authors modified a part of their proposed model and excluded the part where this particular model appears from Movie2. This modified model seems to work well in the current situation for KIF14.

The author has responded well to other reviewer's points and requests, and has made it more understandable for general reader. I think the revised manuscript is suitable for publication in Nature communication.

We thank the reviewer again for his positive evaluation of the manuscript and for his constructive input.

Reviewer #3 (Remarks to the Author):

The authors have responded reasonably to comments on the previous version of this manuscript. I support publication without further change.

We thank the reviewer for his constructive input and positive evaluation.